# Reciprocal regulation of ARPP-16 by PKA and MAST3 kinases provides a cAMP-regulated switch in protein phosphatase 2A inhibition

Veronica Musante[1], Lu Li[2], Jean Kanyo[3], Tukiet T Lam[3], Christopher M Colangelo[3], Shuk Kei Cheng[4], A Harrison Brody[4], Paul Greengard[4], Nicolas Le Novère[2], Angus C Nairn[1]*

[1]Department of Psychiatry, Yale University School of Medicine, New Haven, United States; [2]The Babraham Institute, Cambridge, United Kingdom; [3]W.M. Keck Biotechnology Resource Laboratory, Yale University School Medicine, New Haven, United states; [4]Laboratory of Molecular and Cellular Neuroscience, The Rockefeller University, New York, United States

**Abstract** ARPP-16, ARPP-19, and ENSA are inhibitors of protein phosphatase PP2A. ARPP-19 and ENSA phosphorylated by Greatwall kinase inhibit PP2A during mitosis. ARPP-16 is expressed in striatal neurons where basal phosphorylation by MAST3 kinase inhibits PP2A and regulates key components of striatal signaling. The ARPP-16/19 proteins were discovered as substrates for PKA, but the function of PKA phosphorylation is unknown. We find that phosphorylation by PKA or MAST3 mutually suppresses the ability of the other kinase to act on ARPP-16. Phosphorylation by PKA also acts to prevent inhibition of PP2A by ARPP-16 phosphorylated by MAST3. Moreover, PKA phosphorylates MAST3 at multiple sites resulting in its inhibition. Mathematical modeling highlights the role of these three regulatory interactions to create a switch-like response to cAMP. Together, the results suggest a complex antagonistic interplay between the control of ARPP-16 by MAST3 and PKA that creates a mechanism whereby cAMP mediates PP2A disinhibition.

*For correspondence: angus.nairn@yale.edu

**Competing interests:** The authors declare that no competing interests exist.

## Introduction

ARPP-16 and ARPP-19 were originally identified together with DARPP-32 and ARPP-21 as a group of cAMP-regulated phosphoproteins (ARPPs) enriched in striatal neurons (*Walaas et al., 1983*; *Girault et al., 1990*; *Horiuchi et al., 1990*). ARPP-16 and ARPP-19 are alternatively spliced variants, with ARPP-19 containing 16 additional amino acids at its N-terminus. ARPP-16/19 are also related to endosulfine (ENSA or ARPP-19e), a distinct gene product that shares high identity with the common region of ARPP-16/19 but contains a unique N-terminal region (*Horiuchi et al., 1990*; *Peyrollier et al., 1996*; *Heron et al., 1998*; *Dulubova et al., 2001*). ARPP-16 is present only in the brain and is highly expressed in medium spiny neurons (MSNs) in striatum (*Girault et al., 1990*). In contrast, ARPP-19 is ubiquitously expressed with low levels in striatum (possibly also in non-neuronal cells). ENSA is expressed in all tissues, is widely distributed in the brain, and represents the predominant proportion of the 19 kDa forms of the ARPP-19/ENSA proteins found in striatum (*Dulubova et al., 2001*).

The ARPP-16/19/ENSA proteins have become the focus of many studies in the past few years as a result of their discovery as potent inhibitors of the serine/threonine protein phosphatase, PP2A. PP2A exists in eukaryotic cells as a heterotrimer of a catalytic C subunit, a scaffolding A subunit, and

a variable B subunit that serves to influence the substrate specificity of the phosphatase (*Janssens and Goris, 2001*; *Shi, 2009*). PP2A heterotrimers play critical roles in diverse signaling pathways including regulation of the cell cycle, cell proliferation, and neuronal signalling, and deficits in PP2A function are implicated in many human diseases including cancer, Alzheimer's disease and major depressive disorder (*Sontag and Sontag, 2014*; *Network and Pathway Analysis Subgroup of Psychiatric Genomics Consortium, 2015*; *Sangodkar et al., 2016*). Studies of cell cycle regulation in frog oocytes had found that PP2A including the B55 subunit directly dephosphorylates Cdk substrates upon mitotic exit and that inhibition of PP2A-B55 by a mechanism involving Greatwall (Gwl) kinase was required during mitosis (*Castilho et al., 2009*; *Mochida et al., 2009*; *Vigneron et al., 2009*). Subsequent work revealed that ARPP-19 and ENSA when phosphorylated by Gwl act as critical PP2A inhibitors (*Gharbi-Ayachi et al., 2010*; *Mochida et al., 2010*).

In mammalian cells, MASTL (for microtubule-associated serine/threonine kinase like) is functionally equivalent to Gwl (*Voets and Wolthuis, 2010*), and Gwl/MASTL are now known to play an important role in mitotic progression in several model systems, including Drosophila melanogaster, Xenopus laevis and mammalian cell lines (*Yu et al., 2004*, *2006*; *Archambault et al., 2007*; *Burgess et al., 2010*; *Voets and Wolthuis, 2010*; *Glover, 2012*; *Lorca and Castro, 2012*; *Álvarez-Fernández et al., 2013*; *Wang et al., 2013*). Additional functions have also been reported for ENSA/ARPP-19 proteins. Genetic studies in flies and biochemical studies in Xenopus suggest that ARPP-19 is also involved in meiotic division (*Von Stetina et al., 2008*; *Rangone et al., 2011*; *Wang et al., 2011*; *Dupré et al., 2013*). In budding yeast, inhibition of PP2A-B55 by Igo1/2 (the equivalent of the ARPP proteins) leads to an increased phosphorylation of Gis1, a transcription factor important for cellular quiescence after glucose deprivation (*Bontron et al., 2013*). We have also recently identified ARPP-16 as a PP2A inhibitor in mammalian brain when phosphorylated at Ser46 by MAST3 kinase (*Andrade et al., 2017*), a homolog of MASTL/Gwl, which is active in post-mitotic neurons and is highly expressed in striatum (*Garland et al., 2008*). Conditional knockout of ARPP-16 in mouse forebrain has functional effects on dopaminergic signaling via DARPP-32 and ERK/MAPK, and leads to altered behavioral responses (*Andrade et al., 2017*).

Together, these various studies clearly identify ARPP-16, ARPP-19 and ENSA as important components of cellular signaling in a variety of cell types and organisms. However, several important questions remain to be answered. Our studies in striatal neurons indicate that Ser46 is phosphorylated to a high basal stoichiometry by MAST3 (*Andrade et al., 2017*), which is opposite to that found in non-mitotic phases in dividing cells (*Dupré et al., 2013*, *2014*). Notably, the function of phosphorylation of the ARPP/ENSA proteins by PKA is not known. In striatal cell preparations, activation of PKA leads to increased phosphorylation of Ser88 of ARPP-16, but concomitantly to substantial dephosphorylation at Ser46 of ARPP-16 (and also the equivalent Ser62 of ENSA) (*Andrade et al., 2017*). These results suggest that rather than regulate ARPP-16 activity directly, cAMP-dependent signaling may serve to attenuate phosphorylation of Ser46 leading to disinhibition of PP2A. Related to this possibility, ARPP-19 phosphorylated by PKA was found to be a key element for maintaining prophase arrest (*Dupré et al., 2014*).

In the current study, we investigated the regulatory roles of ARPP-16 phosphorylation by MAST3 and PKA using a combination of biochemical and modeling methods. The results indicate that a mutually antagonistic relationship exists between phosphorylation of Ser46 by MAST3 and Ser88 by PKA that serves as a potential toggle switch mechanism. Moreover, PKA is able to phosphorylate MAST3 leading to its inhibition. The complex interplay likely serves to underlie the balance of signaling pathways that act via ARPP/ENSA and PP2A in distinct ways in either dividing cells or non-dividing neurons.

## Results

### Mutually antagonistic relationship between ARPP-16 phospho-sites Ser46 and Ser88 in vitro

We initially examined in in vitro assays if prior phosphorylation of either Ser46 by MAST3 or Ser88 by PKA influenced the ability of ARPP-16 to be phosphorylated by the alternative kinase. Recombinant ARPP-16 was first phosphorylated at Ser88 by PKA to a maximal stoichiometry and then used as substrate for MAST3 (*Figure 1*). Prior phosphorylation of Ser88 slowed the phosphorylation of

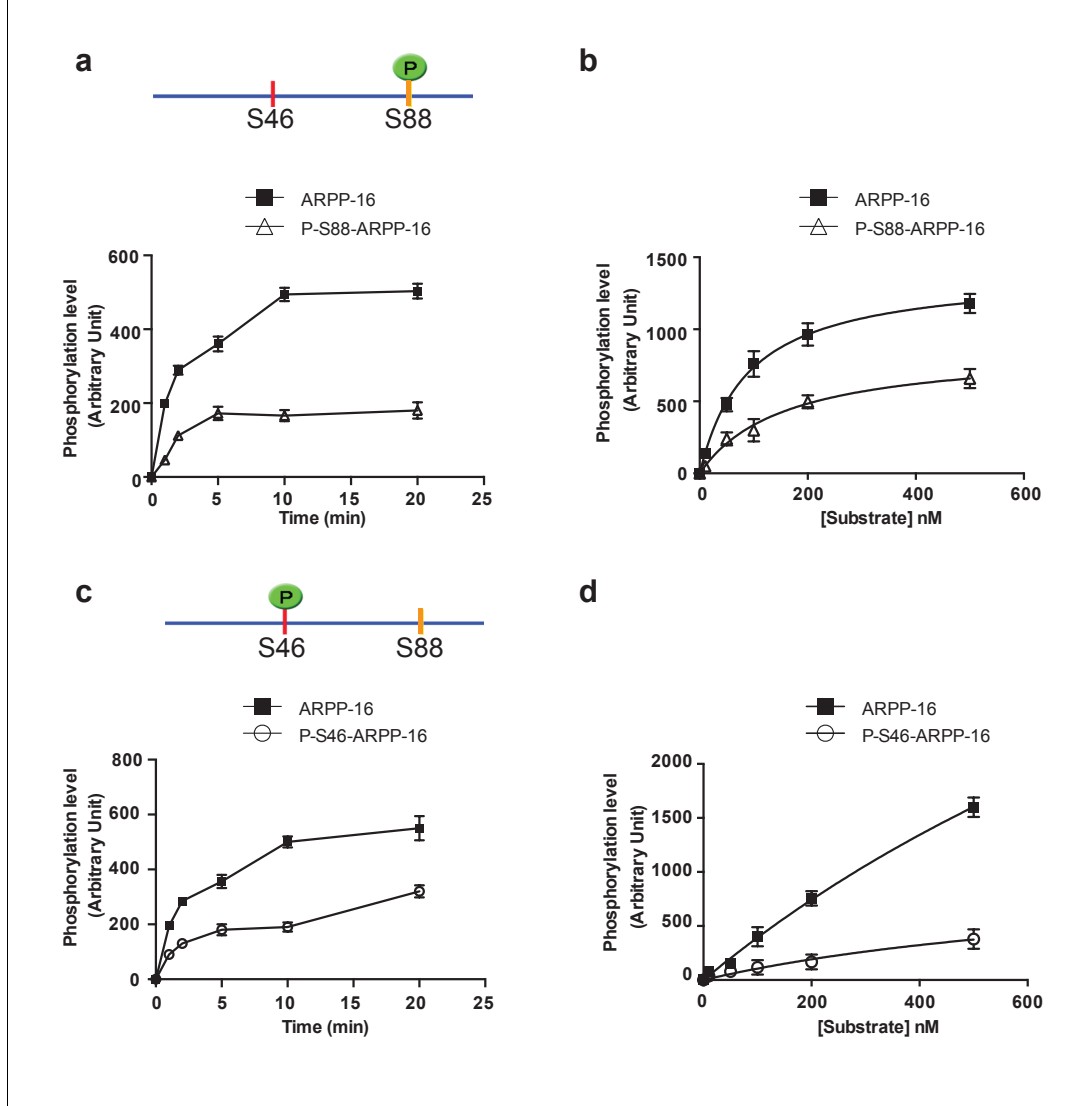

**Figure 1.** PKA-mediated Ser88 phosphorylation attenuates the ability of MAST3 to phosphorylate S46-ARPP-16, and MAST3-mediated phosphorylation of Ser46 attenuates the ability of PKA to phosphorylate S88-ARPP-16. (a) Recombinant purified ARPP-16 or P-S88-ARPP-16 (100 nM) were incubated with ATP-γ-$^{32}$P and MAST3 (overexpressed in HEK293T cells and immunoprecipitated), for various times; proteins were separated by SDS-PAGE and phosphorylation of Ser46 was measured by autoradiography. The resulting values for phosphorylation are expressed in arbitrary densitometric units (a. u.) as mean ± SE of five independent experiments. (b) Assays were carried out as in a. but with increasing concentrations (10–500 nM) of ARPP-16 or P-S88-ARPP-16 for 2 min. Kinetic parameters, determined from double-reciprocal plots, of the data are indicated in *Table 1*. (c) Recombinant purified ARPP-16 or P-S46-ARPP-16 were incubated with ATP-γ-$^{32}$P and PKA for various times; proteins were analyzed as described in panel a. The resulting values for phosphorylation are expressed in arbitrary densitometric units (a.u.) as mean ± SE of five independent experiments. (d) Assays were carried out as in c. with increasing concentrations (10–500 nM) of ARPP-16 or P-S46-ARPP-16. Kinetic parameters, determined from double-reciprocal plots, of the data are presented in *Table 1*.

The following figure supplement is available for figure 1:

**Figure supplement 1.** A phospho-mimetic mutation of S88-ARPP-16 attenuates the ability of MAST3 to phosphorylate S46-ARPP-16.

Ser46 by MAST3 (*Figure 1a*). Kinetic analysis indicated that the prior phosphorylation at Ser88 reduced the enzyme Vmax and increased the Km, effectively reducing the 'catalytic efficiency' by a factor of 3 (*Figure 1b* and *Table 1*). A similar reduction in the efficiency of phosphorylation of Ser46 by MAST3 was observed when a phospho-mimetic version or ARPP-16, S88D-ARPP-16, was used as a substrate (*Figure 1—figure supplement 1*). We also performed the converse analysis where

**Table 1.** Summary of kinetic analysis of ARPP-16 phosphorylation by MAST3 and PKA.

| | MAST3 | Vm<br>$^{32}$P-incorporation/minute | Km<br>nM | Catalytic efficiency<br>Vm/Km |
|---|---|---|---|---|
| *Figure 1b* | ARPP-16 | 1388 ± 27 | 90 ± 5 | 15.4 |
| | P-S88-ARPP-16 | 866 ± 59 | 158 ± 26 | 5.5 |
| *Figure 1—figure supplement 1* | ARPP-16 | 1391 ± 34 | 92 ± 6 | 15.1 |
| | P-S88D-ARPP-16 | 465 ± 32 | 138 ± 24 | 3.4 |
| | PKA | Vm | Km | Catalytic efficiency |
| | | $^{32}$P-incorporation/minute | nM | Vm/Km |
| *Figure 1d* | ARPP-16 | 6999 ± 127 | 1685 ± 51 | 4.1 |
| | P-S46-ARPP-16 | 1027 ± 39 | 870 ± 47 | 1.2 |

ARPP-16 was first maximally phosphorylated by MAST3 at Ser46 and then used as substrate for PKA (*Figure 1c,d*). The prior phosphorylation of Ser46 resulted in slower phosphorylation of Ser88 by PKA, which was reflected in ~7 fold reduction in Vmax, although notably there was a ~2 fold decrease in Km. (*Table 1*). There was no difference in the ability of PKA to phosphorylate Ser88 in S46D-ARPP-16 (data not shown), indicating that the S46D mutation did not act as a phospho-mimetic.

## cAMP-dependent regulation of ARPP-16 phosphorylation on Ser46 and Ser88 in intact cells

ARPP-16 is only highly expressed in striatum, where its levels are low at birth and continue to rise postnatally for several weeks (*Girault et al., 1990*). MAST3 is also selectively expressed in striatum (*Garland et al., 2008*). To further investigate the regulation of ARPP-16 phosphorylation in intact cells, we considered the use of various preparations including acutely isolated striatal slices, primary cultures enriched for striatal neurons, or cell culture. Acutely prepared striatal slices are an ex vivo preparation that are useful for studies that involve pharmacological manipulation (*Andrade et al., 2017*) but are not easily amenable to molecular manipulation. In primary neuronal cultures, there was no detectable expression of ARPP-16 and only low levels of expression of ENSA compared to that found in adult striatum (not shown). Similarly, the expression of MAST3 was very low in cultured striatal neurons compared to adult striatum (not shown).

We therefore decided to generate a cell-based model using HEK293T cells in which ARPP-16 and MAST3 were expressed at levels similar to that found in adult striatum. Using the HEK293T cell model, we first investigated the interaction of the Ser46 and Ser88 phosphorylation sites. Under basal conditions, the phosphorylation of Ser46 was low (*Figure 2*, panel a, lane 1) reflecting low activity of endogenous MAST-related kinases. The phosphorylation of Ser88 was also very low reflecting low basal cAMP levels (lane 1). Forskolin treatment led to a marked increase in phosphory-lation of Ser88 (lane 2). Expression of HA-MAST3 led to a marked increase in phosphorylation of Ser46 (lane 3). However, the phosphorylation of Ser46 in the presence of MAST3 expression was sig-nificantly reduced by forskolin treatment compared to untreated, but MAST3-expressing cells (lane 4). There was a small but not statistically significant effect of MAST3 expression on the level of phos-phorylation of Ser88 following forskolin treatment (lane 4). We also expressed separately S46D-ARPP-16 and S88D-ARPP-16. The increase in phosphorylation of Ser88 observed in the presence of forskolin was unaffected in the S46D-ARPP-16 mutant (lanes 5 and 6). However, there was no increase in phosphorylation of Ser46 in S88D-ARPP-16 when MAST3 was over-expressed (lanes 7 and 8). These results are consistent with the in vitro studies shown in *Figure 1*, and together, indi-cate that PKA-mediated phosphorylation of Ser88 suppresses Ser46-phosphorylation by MAST3,

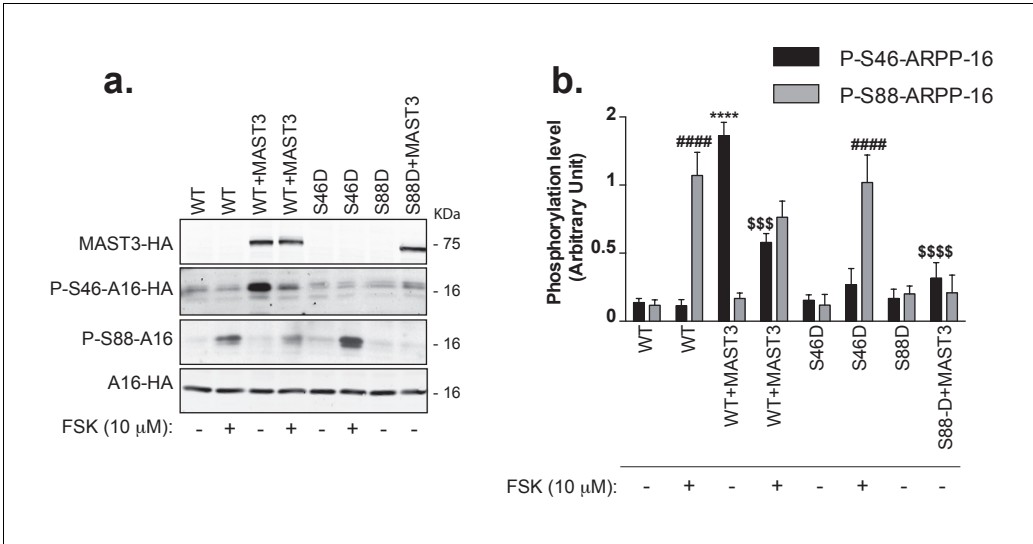

**Figure 2.** cAMP signaling increases phosphorylation of Ser88 and decreases phosphorylation of Ser46 in HEK293T cells. (a) ARPP-16-HA (WT) or the phosphomutants S46D-ARPP-16 or S88D-ARPP-16 were expressed in HEK293T cells alone or in the presence of MAST3-HA kinase. Cells were incubated without or with 10 μM forskolin (FSK) for 30 min. Levels of phosphorylation for Ser46 and Ser88 were measured by immunoblotting with phospho-specific antibodies on SDS-PAGE-resolved cell lysates. Phospho-site signals were normalized for total ARPP-16-HA expression assayed by immunoblotting using anti-HA antibody (A16–HA). For P-S88-ARPP-16 the signal for total ARPP-16-HA was quantified from a separate blot of the same samples. (b) Graph of summary data shows phosphorylation at the different sites expressed in arbitrary densitometric units (a.u.) as mean ± SE of six independent experiments, and analyzed using a one-way ANOVA, multiple comparison test (post-hoc test Tukey). For P-S46-ARPP-16: ****$p<0.001$, ARPP-16/MAST3 vs ARPP-16 ctrl; $$$ $p<0.005$, ARPP-16/MAST3 vs ARPP-16/MAST3/FSK; $$$$$p<0.001$ ARPP-16/MAST3 vs ARPP-16-S88D/MAST3. For P-S88-ARPP-16: ####$p<0.001$, ARPP-16/FSK vs ARPP-16 ctrl; ####$p<0.001$, S46D-ARPP-16/FSK vs ARPP-16 ctrl.

The following figure supplement is available for figure 2:

**Figure supplement 1.** Regulation of ARPP-16 phosphorylation by the D1 receptor agonist, SKF-81297, in striatal slices.

while MAST3-mediated phosphorylation of Ser46 suppresses PKA-mediated phosphorylation of Ser88.

## Mathematical modeling supports a switch-like inhibition of PP2A by ARPP-16

To investigate whether mutual inhibition between the two phosphorylation sites on ARPP-16 is capable of producing a switch-like response to cAMP level changes, we set up a mathematical model focusing on the reciprocal inhibition of MAST3 and PKA by mutually antagonist phosphorylation sites in ARPP-16 (see Materials and methods and Appendix 1—the mutual inhibition model). Steady-state levels of P-S46 and P-S88 ARPP-16 were studied in a two-variable phase-plane (*Figure 3a*). The ensemble of all the system states where a variable does not change over time form a curve called a nullcline. Whenever the P-S46 nullcline intersects the P-S88 one at a given cAMP concentration, both phosphorylation sites are at steady states. As a result of the mutual inhibition, high phosphorylation of Ser46 was associated with low phosphorylation on Ser88 and vice versa (P-S46 nullcline, blue curve). However, the shape of the P-S88 nullcline (red curves) depends on cAMP concentration. An increase of cAMP concentration activates PKA and shifts the nullcline of P-S88 to the right. At low cAMP, there is only one stable steady state corresponding to the full phosphorylation at Ser46 and close to zero phosphorylation at Ser88, as Ser46 phosphorylation suppresses basal PKA activity. At highest cAMP concentration, only one stable steady state exists where ARPP-16 is almost fully

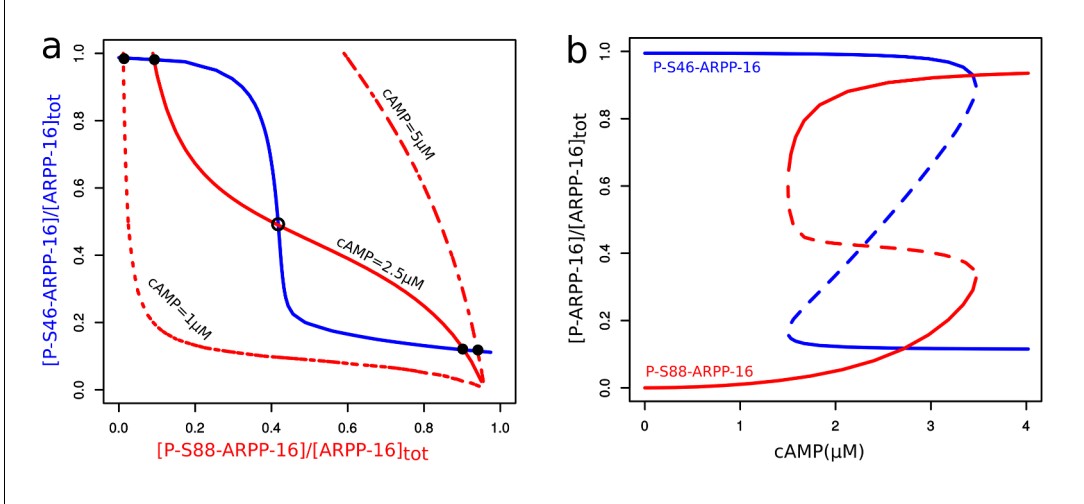

**Figure 3.** Mathematical modeling shows bistability derived from the reciprocal interactions of ARPP-16 phosphorylation sites. (a) Nullclines of P-S46-ARPP-16 and P-S88-ARPP-16. The steady states of P-S46-ARPP-16 (blue) and P-S88-ARPP-16 (red) are plotted as functions of the concentrations of each other. The intersections of the red and blue curve define the steady states of the system at different cAMP concentrations. Filled circles represent stable steady states, whereas the empty circle indicates an unstable state. (b) Antagonism between P-S46-ARPP-16 and P-S88-ARPP-16 creates a switch-like cAMP response. The figure shows bifurcation diagrams of P-S46-ARPP-16 (blue) and P-S88-ARPP-16 (red) plotted as functions of cAMP concentration. Solid lines show stable steady state solutions whereas dashed lines indicate unstable state values.

The following figure supplements are available for figure 3:

**Figure supplement 1.** Bifurcation diagrams of P-S46-ARPP-16 under mutual inhibition (blue, layer 1), PKA inhibition of MAST3 (green, this effect is in addition to previous effect, layers 1 + 2), P-S88-ARPP-16 dominant negative effect on P-S46-ARPP-16 inhibiting PP2A (pink, this effect is in addition to previous effects, layers 1 + 2 + 3), and mutual inhibition with P-S88-ARPP-16 dominant negative effect (black, layers 1 + 3).

**Figure supplement 2.** Bifurcation diagrams of inhibited PP2A under three different layers of regulation described in *Figure 3—figure supplement 1*.

**Figure supplement 3.** Bifurcation diagrams of P-S46-ARPP-16 under different total MAST3 concentrations.

**Figure supplement 4.** Bifurcation diagram of P-S46-ARPP-16, in terms of total MAST3 concentration changes at a fixed cAMP concentration (1.5 μM).

**Figure supplement 5.** Double-parameter bifurcation diagram (cAMP and MAST3tot) showing how total concentration of MAST3 affects the two cAMP thresholds (LP: limit points shown in *Figure 3—figure supplement 3*); and how cAMP concentration affects the two MASTtot thresholds.

**Figure supplement 6.** Bifurcation diagrams of P-S46-ARPP-16, with increasing P-S46-ARPP-16 inhibitory effect on Ser88 phosphorylation (k46).

**Figure supplement 7.** Double-parameter bifurcation diagram (cAMP and k46) showing how inhibitory effect of P-S46-ARPP-16 affects the two cAMP thresholds.

phosphorylated at Ser88 with only sub-micromolar phosphorylation at Ser46. A bistable situation arises at intermediate cAMP concentrations, the two stable states co-existing with an unstable one.

Bifurcation plots of both Ser88 and Ser46 in response to cAMP also showed two qualitatively different states that overlap at intermediate levels of cAMP, indicating bistability within this range of cAMP concentration (*Figure 3b*). When cAMP concentration increases to more than 3.5 μM, Ser46 phosphorylation abruptly drops whereas Ser88 phosphorylation suddenly increases. In the opposite direction, when cAMP decreases the switch occurs below 1.5 μM creating a hysteresis. Based on the mutually antagonistic relationship between the phosphorylation sites on ARPP-16, our model predicts a switch-like behavior of their phosphorylation states in response to changes in cAMP level.

This switch-like response to cAMP concentration change was also detected in experiments where mouse striatal brain slices, an ex vivo model, were used to assess the effect of stimulation of PKA in

a more physiological context. In striatal slices, ARPP-16 and ENSA were phosphorylated at high basal levels by MAST3 (*Figure 2—figure supplement 1*). Stimulation with SKF81297 (a D1 receptor agonist), like forskolin, led to an increase in Ser88 phosphorylation which was accompanied by decreased phosphorylation of Ser46.

## PKA phosphorylates and regulates MAST3 kinase

The above studies with HEK293T cells and the in vitro assays indicate that MAST3 is basally active. We next investigated if PKA might be able to phosphorylate and regulate MAST3. MAST3-HA was expressed in HEK293T cells and isolated by immunoprecipitation. MAST3-HA was incubated with PKA in the presence of $^{32}$P-ATP. PKA phosphorylated MAST3 rapidly reached a plateau after ~10 min (*Figure 4a*). We then examined the effect of PKA-dependent phosphorylation on MAST3 activity using ARPP-16 as substrate for MAST3. PKA-phosphorylated MAST3 was much less active than control MAST3 (*Figure 4b*).

To investigate the regulation of MAST3 further, and identify directly sites of phosphorylation, we expressed MAST3-HA in HEK293T cells and incubated them without or with forskolin. MAST3-HA was then immunoprecipitated, samples digested, phospho-peptides enriched with $TiO_2$, and peptides identified by LC-MS/MS (*Supplementary file 1*). A total of 12 phospho-sites were identified under control conditions. Of these, 6 were found also in the forskolin samples, while an additional 7 phospho-sites were found only in the samples incubated with forskolin. Notably, inspection of the amino acid sequence of MAST3 using bioinformatics tools KinasePhos (*Huang et al., 2005*), PhosphoNet (*Safaei et al., 2011*), NetPhos 3.1 (*Blom et al., 2004*) also identified Thr389 (RHRDT[389] RQR) as a potential PKA site. Although MS/MS analysis of a chymotryptic digest of MAST3-HA showed the presence of a phospho-peptide containing T389 the level of this peptide was very low and we did not observe the presence of this phosphorylated site in any tryptic digestion. We then assessed the effects of forskolin on the phosphorylation of Ser512, Thr628 and Ser747 in different experiments. We used a label-free approach to reanalyze some of the initial samples, and found a 4.8-fold increase in Ser512, 47-fold increase in Thr628, and 15-fold increase in Ser747 (*Figure 4c*). In a second set of experiments, the data-independent SWATH MS/MS technique was used to quantify the effect of incubation with forskolin. An increase in phosphorylation of Ser512, Thr628 and Ser747 of 1.2–2.5 fold was found (average of two separate experiments). Finally, using a label-free approach in a new set of samples, increases of 1.6-fold in Thr628, and 7.1-fold in Ser747 were observed, although Ser512 was recovered in low amounts and no significant changes were observed. Together, the results suggest that MAST3 is phosphorylated at multiple sites by PKA.

## cAMP/PKA-dependent regulation of MAST3 kinase in intact cells

We investigated the potential role of phosphorylation of four sites, Thr389, Ser512, Thr628 and Ser747 through expression of WT MAST3-HA, and mutants in which each site was mutated to either alanine or aspartate to mimic non-phosphorylatable or potential phosphomimetic forms, respectively. We focused first on the effect of Thr389 phosphorylation. We immunoprecipitated WT, T389D-MAST3 and T389A-MAST3 from HEK293T cells, and carried out in vitro assays to directly assess the effects of PKA-dependent phosphorylation on MAST3 activity (*Figure 5a*). Equivalent levels of MAST3 were recovered by immunoprecipitation. However, the T389D-MAST3-HA mutant was much less active than WT-MAST3-HA. In contrast, even after pre-incubation with PKA and ATP, the T389A-MAST3-HA mutant was only slightly less active than WT MAST3-HA. We next expressed ARPP-16 in HEK293T cells in combination with WT-, T389D- or T389A-MAST3 and incubated without or with forskolin (*Figure 5b,c*). As shown in *Figure 2* for WT MAST3-HA, forskolin treatment led to an increase in Ser88 phosphorylation and a decrease in Ser46 phosphorylation. Cells expressing T389D-MAST3 exhibited a large reduction in the phosphorylation of Ser46 compared to WT-MAST3. In cells expressing T389A-MAST3 in the absence of forskolin, phosphorylation of Ser46 was similar to WT MAST3-HA. In comparison to WT MAST3-HA, addition of forskolin resulted in a smaller reduction in Ser46 phosphorylation. Phosphorylation of Ser88 was increased following incubation with forskolin for either WT or T389A-MAST3-HA.

Phosphorylation of Ser46 was reduced for the S512D mutant but not as dramatically as for the T389D mutant (*Figure 6a,b*). For the S512A mutant the effect of forskolin on Ser46 phosphorylation was similar to that observed for WT MAST3-HA. Ser88 phosphorylation was stimulated by forskolin

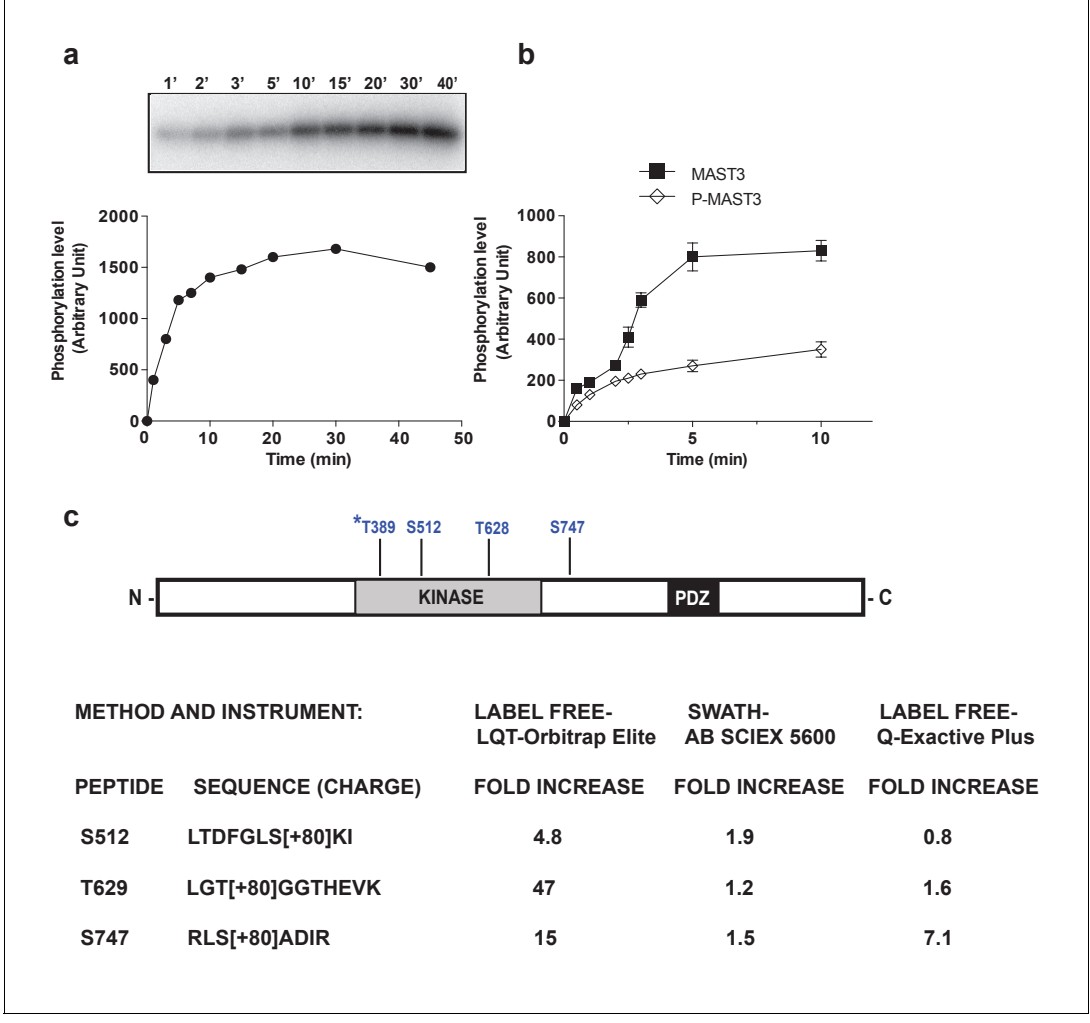

**Figure 4.** MAST3 phosphorylation by PKA in vitro inhibits MAST3 kinase activity; summary scheme of mass spectrometry results showing phosphorylation sites in MAST3. (a) MAST3-HA kinase (overexpressed in HEK293T cells and immunoprecipitated), was incubated with ATP-γ-$^{32}$P and PKA for various times; proteins were separated by SDS-PAGE and phosphorylation of MAST3-HA was measured by autoradiography (upper panel). Summary data (lower panel) are expressed in arbitrary densitometric units (a.u.) as mean ± SE of five independent experiments. (b) MAST3-HA was pre-incubated without (MAST3) or with PKA (P-MAST3) and ATP for 30 min. Recombinant purified ARPP-16 (100 nM) was incubated with P-MAST3 or MAST3 in the presence of ATP-γ-$^{32}$P, for various times; proteins were analyzed as described in panel a. The resulting values for phosphorylation are expressed in arbitrary densitometric units (a.u.) as mean ± SE of three independent experiments. (c) The domain structure of MAST3 is illustrated. The position of the four phosphorylation sites studied are indicated. MAST3-HA was overexpressed in HEK293T cells and incubated in the absence or presence of 10 µM forskolin (FSK) for 30 min. MAST3-HA was isolated by immunoprecipitation and the samples analyzed by LC-MS/MS. The fold-change increase in phosphorylation of S512, T628 and S747 was assessed by different proteomic methods, namely LABEL FREE or SWATH, on different mass spectrometers. Data are presented as –fold change in peptide phosphorylation in response to forskolin compared to control. *T389: Phosphorylation of T389 was identified but not quantitated.

to a similar level for the S512A mutant. For the T628 mutants, phosphorylation of Ser46 was low for the T628D mutant (*Figure 6c,d*) but not as low as for the T389D mutant. Addition of forskolin resulted in a reduction in Ser46 phosphorylation to a level comparable to WT MAST3-HA. Phosphorylation of Ser88 was increased following incubation with forskolin for either WT or T628A-MAST3-HA. In contrast, mutation of Ser747 to aspartate had no significant effect on Ser46 phosphorylation (*Figure 6c,d*). Moreover, mutation to alanine did not prevent the ability of forskolin to reduce phosphorylation of Ser46.

Taken together, these results supported the conclusion that MAST3 is regulated by phosphorylation at three of the four sites analyzed, namely Thr389, Ser512 and Thr628. The in vitro and intact

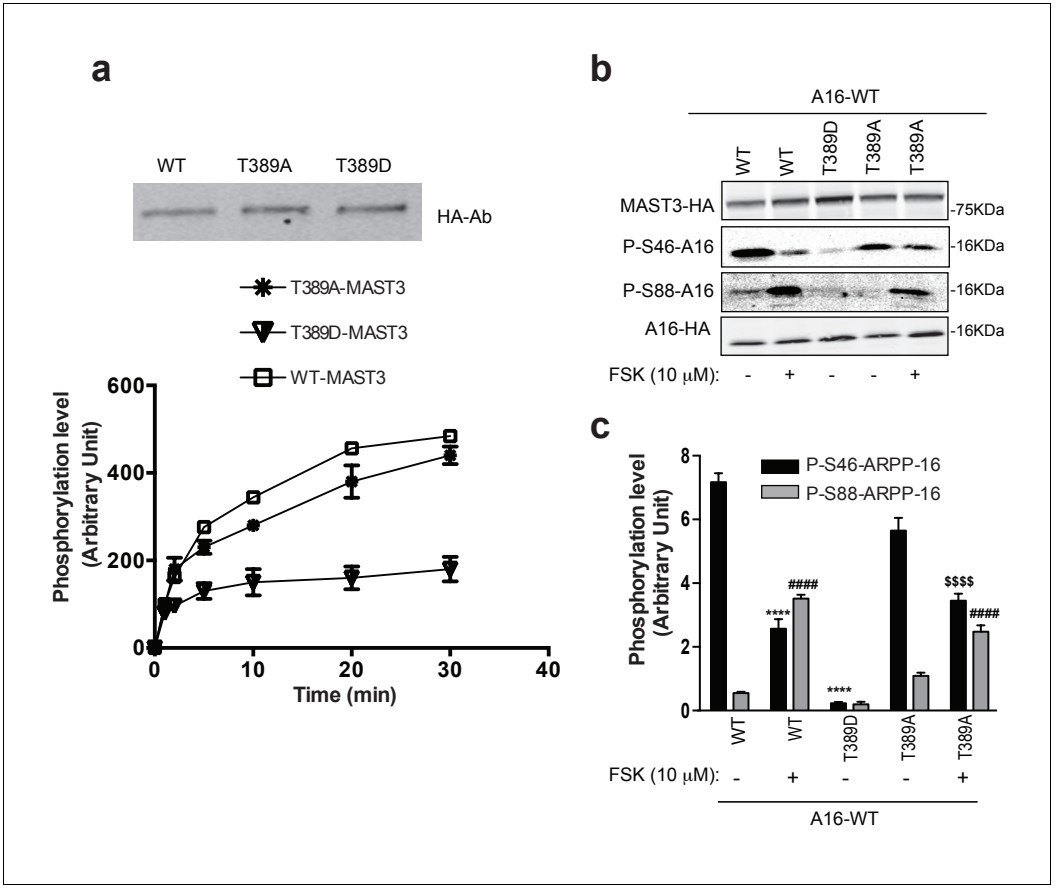

**Figure 5.** Phosphorylation at Thr389 by PKA inhibits MAST3 activity. (**a**) WT MAST3-HA and the unphosphorylable T389A-MAST3-HA or the phosphomimetic T389D-MAST3-HA were expressed in HEK293T cells and isolated by immunoprecipitation. Upper panel shows immunoblotting of WT and mutant MAST-HA proteins. T389A-MAST3 was pre-incubated for 20 min with PKA prior to the assay with ARPP-16. All MAST3 proteins were then incubated with ARPP-16 in the presence of ATP-γ-$^{32}$P for different times. The proteins were resolved by SDS-PAGE and phosphorylation of Ser46 was measured by autoradiography. The results are expressed in arbitrary densitometric units (a.u.) as mean ± SE of three independent experiments. (**b**) ARPP-16-HA was expressed in HEK293T cells alone or with WT MAST3-HA and the phospho-mutants T389D orT389A-MAST3-HA. Cells were treated without or with 10 μM forskolin (FSK) for 30 min. Phosphorylation at Ser46 or Ser88 ARPP-16 were measured by immunoblotting with phospho-specific antibodies. Phospho-site signals were normalized for total ARPP-16-HA expression and then to MAST3-HA proteins, each measured by immunoblotting. (**c**) Graph of summary data shows quantification of phosphorylation on P-S46- and P-S88-ARPP-16 sites expressed in arbitrary densitometric units (a.u.) as mean ± SE of three independent experiments. A one-way ANOVA, multiple comparison test (post-hoc test Tukey) was used for data analysis. P-S46-ARPP-16: ****$p<0.001$ ARPP-16/MAST3 vs ARPP-16/MAST3/FSK; ****$p<0.001$, ARPP-16/MAST3 vs ARPP-16/MAST3-T389D; $$$$ $p<0.001$, ARPP-16/MAST3-T389A vs ARPP-16/MAST3-T389A/FSK. P-S88-ARPP-16: ####$p<0.001$, ARPP-16/MAST3 vs ARPP-16/MAST3/FSK, ####$p<0.001$, ARPP-16/MAST3 vs ARPP-16/MAST3-T389A/FSK. The effect of FSK on MAST3 activity was significantly greater than that of the effect of FSK on T389A-MAST3A, two-tailed T test $p=0.0146$ (not shown).

cell experiments indicated that phosphorylation of Thr389 had the most robust effect on MAST3 activity. Mutation to alanine largely prevented the effect of PKA in the in vitro assay and mutation to a phospho-mimetic residue decreased phosphorylation of ARPP-16 to a very low level at Ser46 in the intact cell experiments. In the intact cell experiments, mutation of Ser512 or Thr628 to aspartate had lesser effects on Ser46 phosphorylation than the T389D mutation. Moreover, the Ser512 or Thr628 to alanine mutants had no effect, presumably because of the predominant effect of Thr389 phosphorylation.

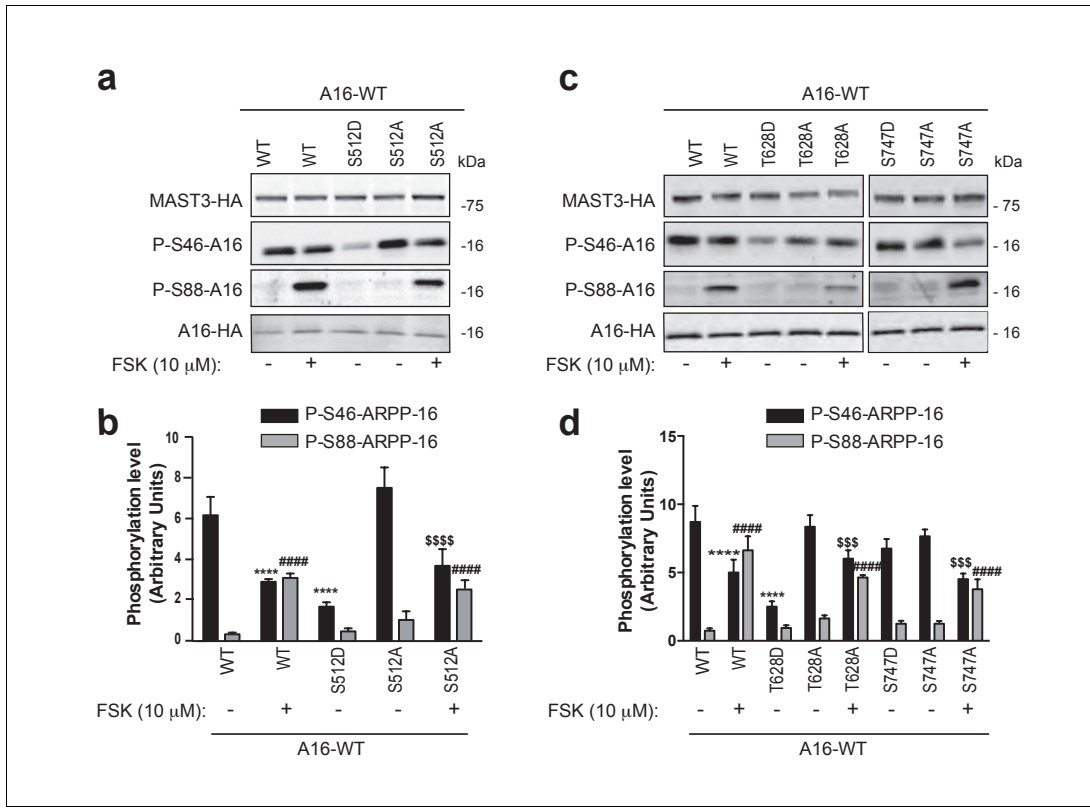

**Figure 6.** Phosphorylation of S512 and T628 but not S747 regulate MAST3 activity. ARPP-16-HA was expressed in HEK293T cells with WT MAST3-HA or the mutants S512D- or S512A-MAST3-HA, T628D- or T628A-MAST3-HA, or S747D- or S747A-MAST3-HA. Cells were incubated without or with 10 μM forskolin (FSK) for 30 min. (a and c) Phosphorylation at Ser46 and Ser88 were measured by immunoblotting. Phospho-site signals were normalized for total ARPP-HA and then to MAST3-HA proteins. (b and d) Graphs of summary data show phosphorylation on the different sites expressed in arbitrary densitometric units (a.u.) as mean ± SE of four (S512) or five independent experiments (T628, S747), one-way ANOVA, multiple comparison test (post-hoc test Tukey). Panel b: P-S46-ARPP-16: ****p<0.001, ARPP-16/MAST3 vs ARPP-16/MAST3/FSK, ****p<0.001, ARPP-16/MAST3 vs ARPP-16/MAST3-S512D; $$$$p<0.001, ARPP-16/MAST3-S512A vs ARPP-16/MAST3-S512A/FSK. Panel b: P-S88-ARPP-16: ####p<0.001, ARPP-16/MAST3 vs ARPP-16/MAST3/FSK, ####p<0.001, ARPP-16/MAST3 vs ARPP-16/MAST3-S512A/FSK. Panel d: P-S46-ARPP-16: ****p<0.001, ARPP-16/MAST3 vs ARPP-16/MAST3/FSK; ****p<0.001 ARPP-16/MAST3-T628D vs ARPP-16/MAST3; $$$p<0.005 ARPP-16/MAST3-T628A vs ARPP-16/MAST3-T628A/FSK; $$p<0.005 ARPP-16/MAST3-S747A vs ARPP-16/MAST3-S747A/FSK. Panel d: P-S88-ARPP-16: ####p<0.001, ARPP-16/MAST3 vs ARPP-16/MAST3/FSK; ####p<0.001, ARPP-16/MAST3 vs ARPP-16/MAST3-T628A/FSK; ####p<0.001, ARPP-16/MAST3 vs ARPP-16/MAST3-S747A/FSK.

We then studied how this direct PKA regulation of MAST3 might affect the switch control over PP2A activity, by extending the mathematical model and examining the dynamic behaviour (see Materials and methods and Appendix 1—the mutual inhibition plus PKA inhibits MAST3 model). The P-S46-ARPP-16 bifurcation diagram indicated that the cAMP concentration range for bistability becomes narrower and shifts towards lower cAMP levels compared with the mutual antagonism scenario alone (*Figure 3—figure supplement 1*). In addition, with this regulation, P-S46-ARPP-16 is dephosphorylated to a much lower level at higher cAMP concentration, therefore, the disinhibition of PP2A at a high cAMP level is more potent (*Figure 3—figure supplement 2*).

## Phosphorylation of ARPP-16 at Ser88 influences the inhibition of PP2A by phospho-Ser46

Our previous in vitro studies have shown that phosphorylation of ARPP-16 at Ser88 has no effect on PP2A activity. We further analyzed in vitro the influence of Ser88 phosphorylation on the ability of

P-S46-ARPP-16 to inhibit PP2A. Recombinant ARPP-16 was thiophosphorylated in vitro at Ser46 to avoid dephosphorylation by PP2A (*Gharbi-Ayachi et al., 2010*; *Mochida, 2014*; *Andrade et al., 2017*). The activity of immunoprecipitated PP2A-B55α complex was measured using a phospho-peptide substrate and the malachite green assay method. As shown previously, P-S46-ARPP-16 inhibited PP2A-B55α (*Figure 7*). Addition of an equimolar amount of dephospho-ARPP-16 had no effect on the ability of P-S46-ARPP-16 to inhibit PP2A. However, addition of an equimolar amount of P-S88-ARPP-16 blocked the inhibitory effect of P-S46-ARPP-16. Addition of an equimolar amount of the S88D-ARPP-16 mutant partially attenuated the effect of P-S46-ARPP-16. These results suggest that in addition to antagonizing the ability of MAST3 to phosphorylate ARPP-16 at Ser46, phospho-Ser88-ARPP-16 may act in a 'dominant-negative' manner to antagonize the inhibition of PP2A by phospho-Ser46-ARPP-16.

Using the mathematical model, we asked how this 'dominant-negative' regulation affects the switch-like control over PP2A activity. We first added this regulation directly into the mutual-antagonism model, assuming that P-S88-ARPP-16 weakens the binding between P-S46-ARPP-16 and PP2A (see Materials and methods and Appendix 1—the mutual inhibition plus PKA inhibits MAST3 and dominant negtive model with modifications in red). The bifurcation diagram of P-S46-ARPP-16 showed that by adding this 'dominant-negative' mode of regulation, the cAMP concentration range required for bistability shifts towards higher cAMP levels (*Figure 3—figure supplement 1*). Indeed, by interfering with PP2A binding, P-S88-ARPP-16 prevents P-S46-ARPP-16 dephosphorylation. Therefore, higher cAMP concentrations are required for both switching off and switching on Ser46 phosphorylation. As P-S88-ARPP-16 antagonizes the inhibition of PP2A, and by doing so preventing

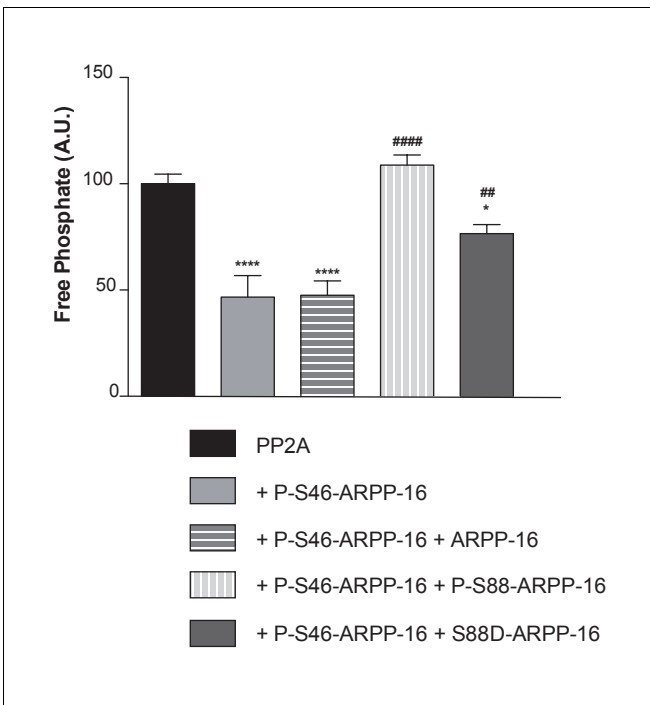

**Figure 7.** ARPP-16 phosphorylation at Ser88 influences the regulation of PP2A activity. Recombinant Flag-Bα was expressed in HEK293T cells and the Bα-PP2A heterotrimer isolated by immunoprecipitation using anti-Flag antibody. PP2A-Bα activity was measured in the presence of thiophosphorylated P-γ-S46-ARPP-16, or a mixture of P-γ-S46-ARPP-16 plus ARPP-16, plus P-S88-ARPP-16, or plus S88D-ARPP-16 (200 nM for each protein). Phosphatase activity was detected using a malachite green assay kit (Millipore). Results are expressed as percent activity with respect to PP2A-Bα activity measured in the absence of any inhibitor (black bar). Data were analyzed by one-way ANOVA multiple comparison test (post-hoc test Tukey) (error bars show SEM). ****$p<0.001$ P-S46-ARPP-16 vs PP2A control; ****$p<0.001$ P-S46-ARPP-16 + ARPP-16 vs PP2A control; ####$p<0.001$, P-S46-ARPP-16 + /P-S88-ARPP-16 vs P-S46-ARPP-16. ##$p<0.01$, P-S46-ARPP-16 + S88D-ARPP-16 vs P-S46-ARPP-16; *$p<0.05$, P-S46-ARPP-16 + S88D-ARPP-16 vs PP2A control.

P-S46-ARPP-16 dephosphorylation, PP2A disinhibition is reduced to a greater extent than the level of Ser46 dephosphorylation (*Figure 3—figure supplement 2*).

We then added the direct regulation of PKA on MAST3 back into the model. The switch-like control of cAMP over PP2A activity was intact. However, combining all three modes of regulation together resulted in only slight modification of PP2A disinhibition, compared to the model with only mutual antagonism and PKA inhibiting MAST3 (*Figure 3—figure supplements 1* and *2*). Based on the data we have and the assumptions we made, this would indicate that the direct regulation from PKA to MAST3 has a much potent effect on controlling PP2A activity than the 'dominant-negative' effect of P-S88-ARPP-16 on PP2A inhibition.

## Discussion

Recent studies have identified the ARPP/ENSA proteins as important regulators of protein phosphatase PP2A (*Gharbi-Ayachi et al., 2010*; *Mochida et al., 2010*; *Dupré et al., 2013*; *Andrade et al., 2017*). The ARPP/ENSA proteins are remarkably conserved in plants, fungi, yeast, flies, worms and mammals (*Dulubova et al., 2001*; *Labandera et al., 2015*) suggesting a primal origin and conserved function to control PP2A activity. However, the precise role of the ARPP/ENSA proteins varies in different cellular contexts, and the proteins are subject to variable modes of regulation. In frog oocytes, ARPP-19 is highly phosphorylated at the PKA site (Ser109, the site equivalent to Ser88 in ARPP-16) during prophase arrest while phosphorylation of Ser67 by Gwl is very low (*Dupré et al., 2013*, *2014*). Upon resumption of meiosis, reduced PKA activity and dephosphorylation of Ser109 is required, following which Ser67 phosphorylation by Gwl acts within an amplification mechanism, converting ARPP-19 in to a potent inhibitor of PP2A (*Dupré et al., 2013*, *2014*). In contrast, in non-dividing neurons in striatum, MAST3 activity is basally active leading to high phosphorylation of Ser46 of ARPP-16, and constitutive inhibition of PP2A (*Andrade et al., 2017*). Correspondingly, phosphorylation of Ser88 in ARPP-16 is low in neurons, but in response to cAMP-dependent activation of PKA, Ser88 phosphorylation is increased while Ser46 phosphorylation decreases, leading to dis-inhibition of PP2A (*Andrade et al., 2017*). That the same conserved signaling module is used in such distinct roles, where the signaling mechanism is essentially inverted, suggests a complex series of interactions between the protein kinases and phosphatases that regulate the phosphorylation of ARPP/ENSA proteins. The results from the current in vitro, cell-based, and modelling studies of ARPP-16, indicate that there is a mutually antagonistic relationship between the phosphorylation of the PKA and MAST3 sites that contribute to a switch-like process that likely controls PP2A regulation in distinct ways in dividing and non-dividing cells (*Figure 8*).

In vitro kinetic analyses indicate that prior phosphorylation of either Ser46 or Ser88 in ARPP-16 acts to suppress the phosphorylation of the other site by its respective kinase, MAST3 or PKA. Studies in intact cells were consistent with the in vitro results. Mutation of Ser88 to aspartate acted as a phospho-mimetic, generating a substrate that was phosphorylated ~5 fold less efficiently by MAST3 in vitro and in intact cells. In contrast, mutation of Ser46 to aspartate did not create a phospho-mimetic in terms of phosphorylation of Ser88 by PKA, consistent with the fact that a Ser46 to aspartate mutation in ARPP-16 failed to inhibit PP2A (Musante unpublished results). The in vitro kinetic assays revealed that prior phosphorylation of either Ser88 or Ser46 influenced both the Km and Vmax parameters by MAST3 or PKA, respectively. ARPP-16, ARPP-19 and ENSA, like DARPP-32 and ARPP-21 are heat-stable proteins that contain little secondary structure and are likely to be highly flexible (*Huang et al., 2001*; *Dancheck et al., 2008*). Presumably, phosphorylation of either Ser46 or Ser88 in ARPP-16 has the ability to constrain the structure(s) of the protein to negatively influence the binding that is reflected in the altered kinetic parameters measured for MAST3 or PKA. Interestingly, injection of a S109D-ARPP-19 protein into intact frog oocytes could not be phosphorylated by Gwl at Ser67 and arrested oocytes in prophase (*Dupré et al., 2014*). This suggests that the mechanism of action of PKA phosphorylation of ARPP/ENSA proteins may be common to both MAST3 and Gwl.

Though initially described as a microtubule-associated protein, little is known about the function or regulation of the MAST family of protein kinases (*Walden and Cowan, 1993*; *Zhou et al., 2004*; *Valiente et al., 2005*; *Wang et al., 2006*). In addition to a conserved kinase domain, MAST1-4 contain a PDZ domain of unknown function, as well as other potential regions that may influence cellular localization or action. In the current study, we found that PKA directly phosphorylates MAST3,

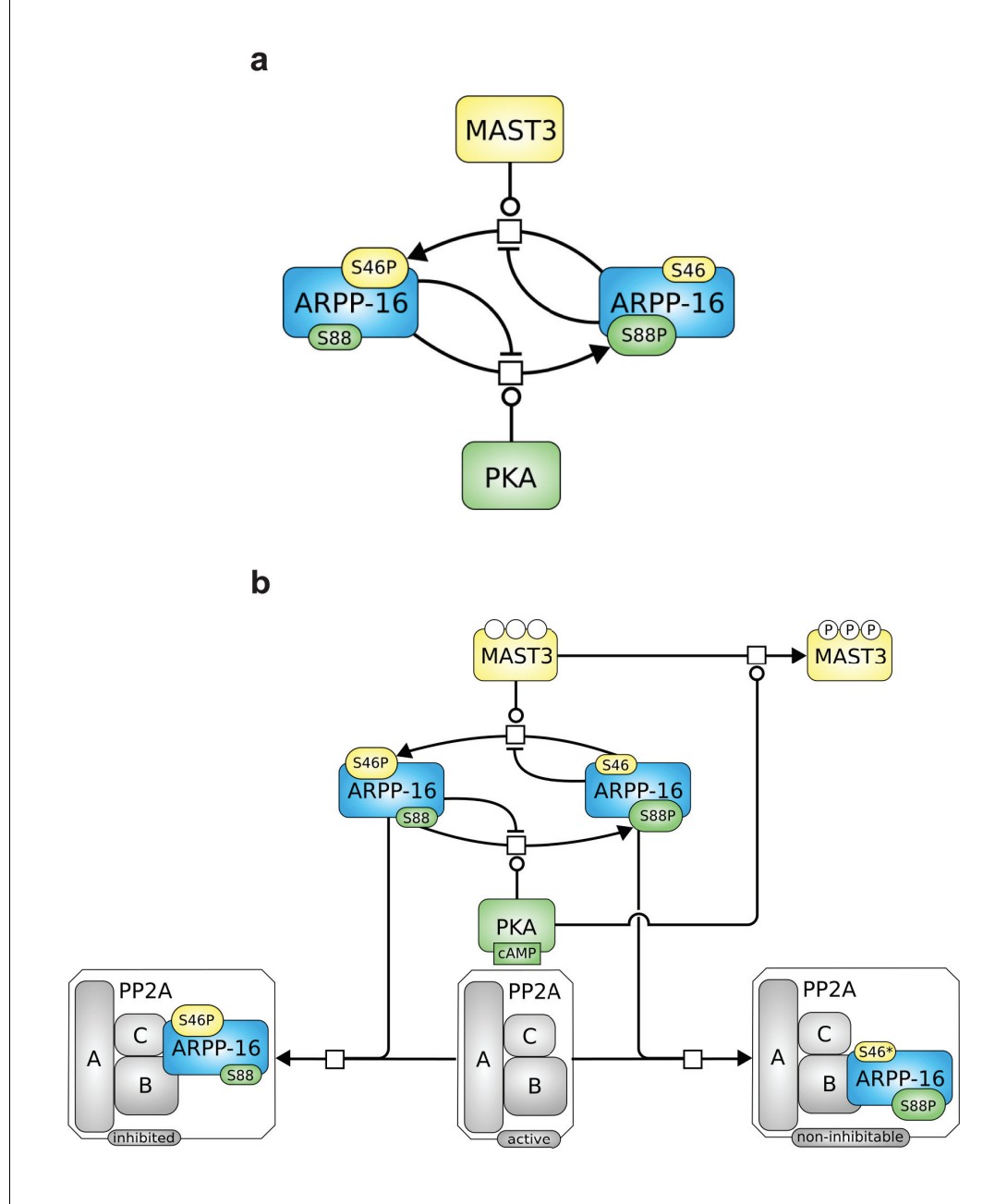

**Figure 8.** Roles of MAST3 and PKA in the regulation of PP2A by ARPP-16. (a) Phosphorylation of ARPP-16 at Ser88 by PKA suppresses the phosphorylation of Ser46 by MAST3, while conversely phosphorylation of Ser46 by MAST3 acts in a reciprocal manner to suppress phosphorylation of Ser88 by PKA. (b) The balance of MAST3 kinase and PKA activities determines the state of phosphorylation of Ser46 and Ser88, and subsequent regulation of PP2A heteromers. SBGN Process Description (*Le Novère et al., 2009*) map showing ARPP-16 phosphorylations by MAST3 and PKA and the different direct and indirect effects of PKA on the modulation of PP2A by ARPP-16. Phosphorylation of ARPP-16 at Ser88 inhibits its phosphorylation at Ser46, phosphorylation of MAST3 inhibits its activity towards S46-ARPP-16, and dominant-negative effect of P-S88-ARPP-16, precluding binding of P-S46-ARPP-16 to PP2A.

leading to inhibition of enzyme activity. Mass spectrometry identified numerous phosphorylation sites including Thr389, Ser512 and Thr628, that contribute to inhibition of MAST3 to different extents, with Thr389 having the largest effect. These 3 sites are all within the kinase domain (residues 367–640 of MAST3) and may influence enzyme activity via direct or allosteric effects.

Thr389 (MAST1, 2, 3, 4), Ser512 (MAST1, 2, 3, 4 and MASTL) and Thr628 (MAST2, 3, 4) are present in other MAST kinase isoforms suggesting conserved modes of regulation of other family members by PKA. In mammals, MASTL kinase is likely widely distributed in dividing cells where it regulates cell cycle progression via the inhibition of ENSA/ARPP-19 (*Voets and Wolthuis, 2010*). Phosphorylation of MASTL by PKA might allow for cAMP-dependent regulation of cell division (see (*Dupré et al., 2014*) for discussion). MAST1-4 exhibit variable patterns of expression in mammalian organs and tissues (*Garland et al., 2008*), where they presumably phosphorylate ENSA/ARPP-19/ ARPP-16, and potentially other substrates. In particular, the different MAST kinase isoforms show distinct patterns of expression in the brain, where they could be regulated by different G protein-coupled receptors that modulate cAMP/PKA signalling (*Gainetdinov et al., 2004*). It will be important to further investigate how MAST3 and the other MAST kinases are regulated by PKA and other phosphorylation events. Currently, there is nothing known about the identity of any of the phosphatase(s) responsible for dephosphorylation of MAST3. Recent studies of Gwl and MASTL have identified specific roles of both PP1 and PP2A in their regulation (*Heim et al., 2015*; *Ma et al., 2016*; *Rogers et al., 2016*; *Wang et al., 2016*). Modeling studies also support a potential role for phosphatase action upon Gwl as part of a negative feedback process that could contribute to its regulation (*Vinod and Novak, 2015*)(see also below). MAST3 and the other MAST kinase isoforms are also likely to be subject to variable modes of regulation at different sites by specific protein phosphatases.

As different MAST kinase isoforms, expressed in distinct patterns in the brain, could regulate the inhibitory effect of ARPP-16/ARPP-19/ENSA on PP2A, we used the model to test how different concentrations of total MAST3 might affect the switch-like control over PP2A inhibition, in response to cAMP concentration changes. Varying total MAST3 concentration in the mathematical model with all three layers of regulation does not affect the general switch-like response to cAMP concentration changes (*Figure 3—figure supplement 3*). In fact, at a fixed cAMP level, bistability exists over a range of MAST3 concentrations (*Figure 3—figure supplement 4*). A two-variable bifurcation diagram reveals wide dynamic ranges for joint MAST3 and cAMP concentrations where bistability exists (*Figure 3—figure supplement 5*).

The structural details of how ARPP-16/ARPP-19/ENSA interact with and inhibit PP2A are not known. ARPP-19/ENSA phosphorylated by Gwl (*Gharbi-Ayachi et al., 2010*; *Mochida et al., 2010*; *Mochida, 2014*) or ARPP-16 phosphorylated by MAST3 (*Andrade et al., 2017*) act as specific inhibitors of heterotrimeric but not AC dimeric forms of PP2A. Studies in Xenopus oocytes suggest a preference for binding of phospho-S67-ENSA to the B55-containing form of the phosphatase (*Castilho et al., 2009*; *Mochida et al., 2010*), while phospho-S46-ARPP-16 can inhibit both B55α and B56δ-containing PP2A heterotrimers (*Andrade et al., 2017*). The interaction of phospho-S46/ S67-ARPP-16/19/ENSA with PP2A likely involves some sort of pseudosubstrate interaction with the active site of the C subunit (*Mochida et al., 2010*; *Williams et al., 2014*). Preliminary structure-function analysis and crosslinking studies indicate that a central region of ENSA containing phospho-Ser67 inhibits PP2A and can bind to both the C and B55 subunit (*Mochida, 2014*). Studies of ARPP-16 indicate that it can bind to sub-regions of the A subunit in an unphosphorylated state (*Andrade et al., 2017*). The ARPP/ENSA proteins therefore are likely to be able to interact through a number of different regions with PP2A heterotrimers. Our studies of the PKA-phosphorylated form of ARPP-16 indicates that it could act in a dominant-negative manner and antagonize the inhibitory actions of phospho-S46-ARPP-16 on PP2A. Perhaps the phospho-S88/S88D forms of ARPP-16 bind favourably to the A or B subunits in a way that competes with the phospho-Ser46 form. The ability of the PKA-phosphorylated ARPP-16 to act as a dominant negative might also provide part of the explanation for the studies carried out in intact Xenopus oocytes which showed that increasing amounts of phospho-S109-ARPP-19 resulting from endogenous PKA activity, or introduction of an Ser109 to aspartate phospho-mimetic form of ARPP-19, blocked the effect of phospho-S67-ARPP-19 on progesterone-induced meiotic maturation (*Dupré et al., 2014*).

Bistability and switch-like behaviour have been considered as basic building blocks in molecular biology and explored both computationally and experimentally (*Gardner et al., 2000*; *Kellershohn and Laurent, 2001*; *Cross et al., 2002*). The discontinuous, two-way switch is often referred to as a toggle switch, in which a system can be switched on or off based on quantitatively different signal thresholds, therefore preventing premature activation or inactivation (*Tyson et al., 2003*). The ARPP/ENSA/Gwl system has been shown to produce a toggle switch response to Cdk1

activity, based on the antagonism between Cdk1 and PP2A (*Mochida et al., 2010*; *Vinod and Novak, 2015*). However, there is so far no model that explores the antagonistic effect between the two phosphorylation sites on ENSA/ARPP. To model PP2A inhibition by ARPP-16, we adapted the mechanism proposed by *Vinod and Novak (2015)* for the inhibition by ENSA. In addition, we focused on the reciprocal relationship between phospho-Ser46 and phospho-Ser88 mediated by inhibition of PKA and MAST3, and demonstrated that ARPP-16 is capable of switching its phosphorylation states depending on different cAMP concentration thresholds, therefore dynamically controlling PP2A activity. The additional layers of PKA regulation, including directly inhibiting MAST3 and dominant negative antagonism of PP2A inhibition, reduce and almost equalize the two cAMP thresholds for switching on or off PP2A inhibition. These regulations together produce an ultra-sensitive system in response to cAMP at micromolar-range concentrations and deepen the release of PP2A inhibition. The micromolar-range cAMP concentrations required for bi-stable states provide sensitive thresholds in most subcellular neuronal compartments, given the fast diffusion coefficient of cAMP (*Hempel et al., 1996*; *Chen et al., 1999*), and the likely very high activity of cAMP phosphodiesterase activity in striatal neurons (*Sharma et al., 2015*). However, the network we studied here is relatively small, and the antagonistic inhibition exerted by ARPP-16's MAST3 phosphorylation site to its PKA site could be strengthened by the feedback loop from PP2A inhibition to PKA inhibition, via for example, phospho-Thr75-DARPP-32 (*Bibb et al., 1999*). As shown in the two-parameter bifurcation diagrams (*Figure 3—figure supplements 6* and *7*), the stronger phospho-Ser46 antagonizes Ser88 phosphorylation, the wider the difference between two cAMP thresholds becomes, therefore the more hysteresis appears. Future modeling work will be useful to understand the complex interrelationship between the abundant PKA substrates, ARPP-16/ENSA, DARPP-32 and ARPP-21/RCS, all of which play important roles in striatal neurons (*Walaas et al., 2011*).

Through in vitro analysis, cell-based studies and molecular modeling, the current study supports a complex inter-relationship between the functional effects of the MAST3 and PKA sites in ARPP-16, and the presence of additional regulatory mechanisms that couple changes in cAMP levels to generate a switch-like control over PP2A activity. Elements of the switch-like process are summarized in the two schematic illustrations shown in *Figure 8*. Phosphorylation of ARPP-16 at Ser88 by PKA suppresses the phosphorylation of Ser46 phosphorylation by MAST3, while conversely phosphorylation of Ser46 by MAST3 acts in a reciprocal manner to suppress phosphorylation of Ser88 by PKA (*Figure 8a*). In turn, the balance of MAST3 kinase and PKA activities determines the state of phosphorylation of Ser46 and Ser88. For example, under basal conditions in neurons in striatum, MAST3 activity is high and PKA activity is low, resulting in high levels of phospho-Ser46 and inhibition of specific heterotrimeric forms of PP2A (*Figure 8b*, left panel). Following its activation by cAMP, PKA acts at three different levels to prevent the inhibition of PP2A isoforms. Phosphorylation of Ser88 suppresses the ability of MAST3 to phosphorylate Ser46. PKA also phosphorylates MAST3 at multiple sites to inhibit its activity (*Figure 8b*, upper right panel). Finally, phospho-Ser88-ARPP-16 acts in a dominant-negative manner to prevent inhibition of PP2A by ARPP-16 phosphorylated at Ser46 (*Figure 8b*, lower right panel). The complex interplay likely serves to underlie the balance of signaling pathways that act via ARPP-16 and PP2A in distinct ways in neurons and other cell types. While the current study and the work of others has elucidated many details of the regulation of PP2A by ARPP-16/19/ENSA, several important questions remain. The identity of the phosphatase(s) that dephosphorylate ARPP-16/19/ENSA at the PKA site, as well as of phosphatases that dephosphorylate various sites in MAST3, are not known. Regulation of these and other phosphatases would potentially add further complexity to this signaling hub. Additional structural analyses will also be needed to elucidate the molecular details related to how GwL/MAST-phosphorylated ARPP-16/19/ENSA inhibit PP2A.

## Materials and methods

### Materials

pET28-ARPP-16, pET28-S46D-ARPP-16, pET28-S88D-ARPP-16, and pCMV-HA-ARPP16, pCMV-HA-S88D-ARPP16, pCMV-HA-S46D-ARPP16, pCMV-HA-MAST3 (sequence corresponding from aa313 to aa1021 of the human protein, all the other mutant constructs are variation of the same sequence, aa1-1021), pCMV-HA-MAST3-FL (human MAST3 full length sequence), pCMV-HA-T389D-MAST3,

pCMV-HA-T389A-MAST3, pCMV-HA-S512D-MAST3, pCMV-HA-S512A-MAST3, pCMV-HA-T628A-MAST3, pCMV-HA-T628D-MAST3, pCMV-HA-S747A-MAST3 and CMV-HA-S747D-MAST3 were prepared using standard procedures (see below). Antibodies used for these studies include: anti-rabbit total ARPP-16 (1:1000 G153, [*Horiuchi et al., 1990*]), anti-rabbit P-S46-ARPP-16 (1:1000 RU1102, [*Andrade et al., 2017*]), anti-rabbit P-S88-ARPP-16 (1:500 G446 [*Dulubova et al., 2001*]), anti-HA antibody (catalog #05–904 RRID:AB_11213751; 1:1000 Cell Signaling Technology, Inc), anti-HA antibody (Novus Biological) used for immunoprecipitation, anti-FLAG (catalog #F1804 RRID:AB_262044; Sigma-Aldrich), anti-GAPDH antibody (catalog CB1001-500UG RRID:AB_2107426; 1:3000, EMD Millipore, anti-MAST3 antibody (catalog #BS5790, AB_2661882; 1:1000, Bioworld Technology, Inc), anti-total GluR1 antibody (catalog #04–855 RRID:AB_1977216; 1:1000, EMD Millipore), anti-pS845 on GluR1 antibody (catalog #p1160-845 RRID:AB_2492128; 1:1000, PhosphoSolutions), monoclonal total DARPP-32 (1:5000 6a [*Ouimet et al., 1984*]), anti-rabbit pT34 on DARPP-32 (1:1000 cc500 [*Stipanovich et al., 2008*]). Recombinant PKA was from Millipore). Forskolin and Okadaic Acid were from Abcam. R(+)-SKF-81297 hydrobromide (catalog S179-5MG) was from Sigma-Aldrich.

## Site-directed mutagenesis of ARPP-16 and MAST3

Mutations of ARPP-16 Ser46 and Ser88 and of each potential PKA phosphorylation site for MAST3 were performed using the QuikChange XL site-directed mutagenesis kit according to the manufacturer's instructions (Stratagene). Briefly, 50 ng of pET28-ARPP-16, pCMV-HA ARPP-16, pCMV-HA MAST3 were used as a template with the following mutant nucleotide primers: for ARPP-16, S46A forward primer 5'-tgcagaaagggcaaaagtattttgatgctggggattacaac-3', S46A reverse primer 5'-gttgtaatccccagcatcaaaatactttttgcccttctgca-3', S46D forward primer 5'- aaaagattgcagaaagggcaaaag-tattttgatgatgggggattacaacatgg-3', S46D reverse primer 5'- ccatgttgtaatccccatcatcaaaa-tactttttgccctttctgcaatctttt-3', S88A forward primer 5'- cctccctcagcggaaaccagccctggttgc-3', S88A reverse primer 5'- gcaaccagggctggtttccgctgagggagg-3', S88D forward primer 5'- ctccctcagcggaaac-cagacctggttgctagc-3', S88D reverse primer 5'- gctagcaaccaggtctggtttccgctgagggag-3'; for MAST3, T389A forward primer 5'-gcggcaccgtgacgcccggcagcgctttg-3', T389A reverse primer 5'-caaagcgctgccgggcgtcacggtgccgc-3', T389D forward primer 5'-gtgcggcaccgtgacgaccgg-cagcgctttgcc-3', T389D reverse primer 5'-ggcaaagcgctgccggtcgtcacggtgccgcac-3', S512A forward primer 5'-ggacttcggcctggccaagatcggcct-3', S512A reverse primer 5'-aggccgatcttggccaggccgaagtcc-3', S512D forward primer 5'-cacggacttcggcctggacaagatcggcctcatg-3', S512D reverse primer 5'-cat-gaggccgatcttgtccaggccgaagtccgtg-3', T628A forward primer 5'-accgtctgggcgctggtggcaccc-3', T68A reverse primer 5'-gggtgccaccagcgcccagacggt-3', T628D forward primer 5'-ggaccgtctgggc-gatggtggcacccac-3', T628D reverse primer 5'-gtgggtgccaccatcgcccagacggtcc-3', S747A forward primer 5'-tggccgccggctggctgctgacatccgg-3', S747A reverse primer 5'-ccggatgtcagcagc-cagccggcggcca-3', S747D forward primer 5'-tggccgccggctggatgctgacatccgg-3', S747D reverse primer 5'-ccggatgtcagcatccagccggcggcca-3'.

DNA sequencing identified appropriate clones.

## Expression of recombinant proteins

6xHis-ARPP-16, 6xHis-S46D-ARPP-16 and 6xHis-S88D-ARPP-16 proteins were overexpressed in BL21 (DE3) cells, lysed using sonication in 300 mM NaCl, 50 mM potassium phosphate, 1% Triton X-100 pH 8.0 and immobilized on Profinity IMAC Ni$^{2+}$ charged resin (Bio-Rad, Hercules, CA). The resin was washed 3 times in 300 mM NaCl, 50 mM potassium phosphate, 10 mM imidazole, pH 8.0. Proteins were eluted using a buffer containing 300 mM NaCl, 50 mM potassium phosphate, 250 mM imidazole, pH 8.0. Buffer exchange (150 mM NaCl, 25 mM TrisHCl pH 7.4), used PD10 desalting columns (GE Healthcare Life Science), 1 ml fractions were collected, and protein content was determined using SDS-PAGE.

## HEK293T protein expression

293T/17 [HEK 293T/17] (ATCC CRL11268) were purchased from AMERICAN TYPE CULTURE COLLECTION (ATCC) (lot number 58483269, cell species identity has been confirmed by COI assay (intraspecies) and STR assay (interspecies); cells tested negative for mycoplasma). Cells were cultured to 60–70% confluence in 10% FBS-DMEM. Expression plasmids (HA-ARPP-16, HA-S46D-ARPP-16, HA-S88D-ARPP-16, HA-MAST3 [aa from 332 to 1014], HA-T389D-MAST3, HA-T389A-MAST3,

HA-S512D-MAST3, HA-S512A-MAST3, HA-T628A-MAST3, HA-T628D-MAST3 and HA-S747A-MAST3 were expressed in HEK293T cells in 10 cm plates using Lipofectamine 2000 transfection (Invitrogen). Immunoblotting for ARPP-16 or MAST3 was routinely used to control for levels of protein expression. Expression levels of ARPP-16 plus endogenous ARPP-19, or MAST3, were approximately equal to that of ARPP-16 plus ENSA, or MAST3, in adult striatum. Twenty-four hours after transfection, cells were lysed in a buffer containing 25 mM Tris, pH 8.0, 150 mM NaCl, 0.1% Triton X-100, protease and phosphatase inhibitors cocktails, followed by centrifugation at 16,000 x g, 10 min. Supernatants were subjected to immunoprecipitation or immunoblotting as described below.

## Striatal slice preparation

Male C57BL/6 mice at 6–8 weeks old were purchased from Charles River. All mice used in this study were handled in accordance with the Yale University Animal Care and Use Committee (IACUC) and NIH *Guide for the Care and Use of Laboratory Animals*. Male C57BL/6 mice were sacrificed by decapitation. The brains were rapidly removed and placed in ice-cold, oxygenated Krebs-HCO$_3^-$ buffer (124 mM NaCl, 4 mM KCl, 26 mM NaHCO$_3$, 1.5 mM CaCl$_2$, 1.25 mM KH$_2$PO$_4$, 1.5 mM MgSO$_4$ and 10 mM D-glucose, pH 7.4). Coronal slices (350 µm) were prepared using a vibrating blade microtome (VT1000S, Leica Microsystems, Nussloch, Germany), as described previously (*Nishi et al., 2005*). From each mouse, six striatal slices were dissected from the coronal slices in ice-cold Krebs-HCO$_3^-$ buffer. Each slice was placed in a polypropylene incubation tube with 2 ml fresh Krebs-HCO$_3^-$ buffer containing adenosine deaminase (10 µg/ml). The slices were preincubated at 30°C under constant oxygenation with 95% O$_2$/5% CO$_2$ for 60 min. The buffer was replaced with fresh Krebs-HCO$_3^-$ buffer after 30 min of preincubation. Each slice was treated with drug as specified in each experiment. After drug treatment, slices were transferred to Eppendorf tubes, frozen on dry ice, and stored at –80°C until assayed. Frozen tissue samples were sonicated in boiling 1% sodium dodecyl sulfate (SDS) and boiled for an additional 10 min. Total lysate (equal volume for each sample) was analyzed by SDS-PAGE and immunoblotting as described below.

## Immunoblotting

Cell lysates were separated by SDS-PAGE (Novex 10–20% or 4–20% gradient gels (Invitrogen)) and transferred onto nitrocellulose membranes, 0.2 µm (Bio-Rad). The membranes were blocked for 1 hr at room temperature (5% nonfat dry milk in PBS, 0.1% Tween 20) and immunoblotted overnight at 4°C, with specific antibodies as indicated. Antibody binding was detected using IRDye800-conjugated anti-mouse IgG (catalog #610-102-041 RRID:AB_2614830; 1:10.000; Rockland) or IRDye680-conjugated anti-rabbit IgG (catalog #926–68021 RRID:AB_10706309; 1:10.000 LI-COR, Bioscience). Blots were analyzed and quantified using an Odyssey Infrared Imaging System (LI-COR Bioscience, Image Studio Lite version 3.1, RRID:SCR_013715). Anti-rabbit pS88 (G446), and anti-rabbit pS46 (RU1102) for ARPP-16 were detected using peroxidase-conjugated secondary antibody (catalog #PI-1000 RRID:AB_2336198; 1:300, Vector Laboratories,Inc.) coupled with a chemiluminescence detection system (Pierce, ThermoScientific).

## ARPP-16 in vitro phosphorylation

For Ser46 phosphorylation, purified 6xHis-ARPP-16 proteins (100 µM) were incubated with immunoprecipitated MAST3 kinase in the presence of 200 µM ATP or thio-ATP (Sigma) in phosphorylation assay buffer (50 mM Hepes, pH 7.4, 10 mM MgCl2) at 30°C for different times. For Ser88 phosphorylation, purified 6xHis-ARPP-16 proteins (100 µM) were incubated with recombinant PKA in the presence of 200 µM ATP or thio-ATP in phosphorylation assay buffer (50 mM Hepes, pH 7.4, 10 mM MgCl$_2$, 1 mM EGTA) at 30°C for different times. Phospho-ARPP-16 preparations were isolated (where needed) by ion-exchange chromatography using an FPLC.

## Immunoprecipitation and phosphatase assay

PP2A-Bα subunit was expressed in HEK293T cells and cells lysed as described above. Lysates were incubated with 50 µl (50% slurry) of anti-FLAG conjugated agarose beads for 2 hr at 4°C. Immunocomplexes were washed 3 times in lysis buffer without phosphatase inhibitors and 2 times in PP2A reaction buffer (pNPP Ser/Thr Assay Buffer, Phosphatase Assay Kit, Millipore). The PP2A-Bα trimer immunocomplex was resuspended in 100 µl of PP2A reaction buffer and incubated without or with

thio-phosphorylated P-S46-ARPP-16, or P-S46-ARPP-16 plus: ARPP-16, or P-S88-ARPP-16, or S88D-ARPP-16 (200 nM each) for 10 min at 37°C in the presence of 500 μM phosphopeptide (K-R-pT-I-R-R). Phosphatase activity was measured using a malachite green assay kit (Millipore).

## MAST3 immunoprecipitation and phosphorylation

MAST3-HA, T389A-MAST3-HA, or T389A-MAST3-HA were expressed in HEK293 and cells lysed as described above. Lysates were incubated with 50 μl (50% slurry) of anti-HA conjugated agarose beads for 2 hr at 4°C. Immunocomplexes were washed 3 times in RIPA buffer and 3 times in a buffer without detergents (150 mM MgCl$_2$, 25 mM Tris-HCl pH 7.4) and resuspended in PKA reaction buffer (50 mM Hepes, pH 7.4, 10 mM MgCl$_2$, 1 mM EGTA). The immunoprecipitated kinases were incubated with PKA in the presence of 200 μM [γ−32P] ATP at 30°C for different times. In the experiments that examined the effect of PKA-dependent phosphorylation on MAST3 activity, MAST3 immunocomplexes were incubated without or with PKA for 30 min together with ATP, washed twice with kinase buffer, then incubated with ARPP-16 as described above.

## Quantification and statistical analysis

Immunoblots and $^{32}$P autoradiograms were analyzed by Fiji ImageJ (http://fiji.sc/wiki/index.php/Fiji) or Image Studio (Licor Bioscience, Lincoln, NE, USA). Graphical presentations and statistical analyses were made using Graph Pad Prism (RRID:SCR_002798; Graph Pad Software).

## Mass spectrometry

Samples were processed in three different ways. (1) In initial studies, phosphopeptides were enriched and subsequently analyzed by liquid chromatography-tandem mass spectrometry (LC-MS/MS) analysis using a LTQ Orbitrap Elite instrument. Label-free quantitation was carried out retrospectively using Progenesis QI software; (2) Information Dependent Acquisition (IDA) and SWATH acquisitions were carried using an AB SCIEX Triple TOF$^{TM}$ 5600 instrument; (3) In later experiments where phosphopeptides were not enriched, samples were analyzed using a Q-Exactive Plus. Label-free quantitation was carried out using Progenesis QI software.

(i) Sample preparation: HEK293T cells were transiently transfected with HA-tagged wild-type MAST3 using Lipofectamine 2000 for 24 hr and then either mock treated with DMSO or treated with 10 μM FSK for 30 min. The cells were lysed and MAST3-HA was immunoprecipitated as described above. For (1) and (3), immunoprecipitated MAST3-HA was subjected to SDS-PAGE, and Coomassie stained gel bands corresponding to the MAST3-HA protein were in-gel digested. For (2), samples analysed by SWATH were not subjected to SDS-PAGE. Samples were digested with Lys C, chymotrypsin, LysC/trypsin, or chymotrypsin/trypsin in attempts to maximize coverage of the protein (see *Supplementary file 1*). Digestions were carried out at 1:10 ratio of enzyme:protein at 37°C for 16 hr. In the later experiments (3), samples were digested with trypsin (MS grade Promega; incubation at 37°C overnight), peptides were extracted utilizing an 80% acetonitrile solution containing 0.1% formic acid, and dried. Dried extracted peptides were reconstituted in Buffer A (100% water, 0.1% formic acid, and analyzed).

(ii) Phosphopeptide enrichment: Phosphopeptides present in initial digested samples were enriched using an in-house titanium oxide (TiO$_2$) enrichment method with Glycen TopTips (Glycen Corporation, Columbia, MD [according to the manufacturer's manual]. Briefly, the samples were acidified with 0.5% trifluoroacetic acid (TFA)–50% acetonitrile and loaded onto TopTips (Glygen Corp.), followed by washing three times with 100% acetonitrile, 0.2 M sodium phosphate buffer (pH 7.0), 0.5% TFA, and then 50% acetonitrile. Phosphopeptides were eluted from the TopTip using 28% ammonium hydroxide, dried in a SpeedVac, and then redried from water. Samples were resuspended in 70% formic acid and then immediately diluted to 0.1% TFA for mass spectrometry analysis. Both the flow-through (FT) from the washes and the eluted enriched (EN) fractions were analyzed by high resolution LC-MS/MS on a LTQ-Orbitrap MS system. Quantitation of phosphopeptide levels was carried out retrospectively using Progenesis QI software.

(iii) Mass spectrometry and data analysis: (1) Samples were analyzed by LC-MS/MS using an LTQ Orbitrap Elite equipped with a Waters nanoACQUITY ultra-performance liquid chromatography (UPLC) system using a Waters Symmetry C$_{18}$180 μm x 20 mm trap column and a 1.7 μm (75 μm inner-diameter x 250 mm) nanoACQUITY UPLC column (35°C) for peptide separation. Trapping was

done at 15 μl/min with 99% buffer A (100% water, 0.1% formic acid) for 1 min. Peptide separation was performed at 300 nl/min with buffer A and buffer B (100% $CH_3CN$ containing 0.1% formic acid). A linear gradient (51 min) was run with 5% buffer B at initial conditions, 50% buffer B at 50 min, and 85% buffer B at 51 min, and re-equilibration was carried out for 20 min. Blank injections runs were implemented in between sample injections to ensure no carryover. Mass spectral data were acquired in the Orbitrap using 1 microscan and a maximum inject time of 900 μs followed by data-dependent MS/MS acquisitions in the ion trap (via collision-induced dissociation [CID]) and in the high-energy collision dissociation (HCD) cell. Neutral loss scans were also obtained for 98.0, 49.0, and 32.7 atomic mass units (amu). The data were analyzed using Mascot Distiller and the Mascot search algorithm.

(2) Information Dependent Acquisition (IDA) and SWATH acquisitions were carried out on an AB SCIEX Triple TOF$^{TM}$ 5600 coupled with a Waters nanoACQUITY UPLC system. Digests were loaded onto a Waters Symmetry $C_{18}$180 μm x 20 mm trap column and separated on a nanoACQUITY 1.7 μm BEH300 $C_{18}$ (75 μm x 150 mm) column. Peptide separations were performed at 500 nl/min with Buffer A (0.1% formic acid) and Buffer B ($CH_3CN$ containing 0.1% formic acid). A linear gradient (70 min) was run with 1% buffer B at initial conditions, 35% buffer B at 70 min, 95% buffer B at 70.33 min, and column re-equilibration for 20 min. IDA runs were carried out at high sensitivity with resolution ~16–18K in the MS/MS (w/max of 30 MS/MS per cycle) at 0.05 s/scan and TOFMS scan of 0.25 s. SWATH acquisition was carried out over the mass range of 400–1250 with setting at 26 Da scan window with 1 Da overlap (i.e. 400–425, 424–450, etc.) for a total of 34 SWATH windows per cycle. Acquired LC MS/MS data were analyzed using Analyst (v.2.5) and PeakView (v.2.0, AB SCIEX). Protein identification and site modification assignment utilized MASCOT (Matrix Science) searches; modification sites of interest were manually verified. Skyline (*MacLean et al., 2010*) was utilized to obtain precursor ion quantitation from SWATH data on the various modification sites.

(3) Samples were isolated from HEK293T cells essentially as described above. Four control and four forskolin-treated samples were analyzed, each as a technical duplicate. Samples were analysed using a Q-Exactive Plus (Thermo Fisher Scientific) LC MS/MS system equipped with a Waters nano-Acquity UPLC system, that used a Waters Symmetry $C_{18}$180 μm x 20 mm trap column and a 1.7 μm (75 μm x 250 mm) nanoACQUITY UPLC column (37°C) for peptide separation. Trapping was done at 5 μl/min, 99% Buffer A for 3 min. Peptide separation was performed with a linear gradient over 140 min at a flow rate of 300 nL/min. Precursor mass scans (300 to 1500 m/z range, target value 3E6, maximum ion injection times 45 ms) were acquired and followed by HCD based fragmentation (normalized collision energy 28). A resolution of 70,000 at m/z 200 was used for MS1 scans, and up to 20 dynamically chosen, most abundant, precursor ions were fragmented (isolation window 1.7 m/z). The tandem MS/MS scans were acquired at a resolution of 17,500 at m/z 200 (target value 1E5, maximum ion injection times 100 ms).

Relative changes in phosphopeptide levels were calculated based on the detected abundances of the precursor mass utilizing Progenesis QI (v. 3.0). Raw mass spectral data were first aligned based on their retention time, then peaks are picked based on a 'co-detection' scheme where ion detection was performed once on a single aggregate run, and multiple isotopic forms of the same ion were grouped to provide the abundance of that ion (the precursor mass) across the various runs. The protein level quantitation of MAST3 was then normalized (based on total abundance method) across all runs to correct for experimental (4 biological replicates) or technical (2 technical replicates) variations. The normalization was then attributed to the quantitation of the precursor mass corresponding phosphopeptide identified in previous runs. The differential analysis used to generate the Anova *p* value takes into account the mean difference and the variance and also the sample size. Thus small differences with small variance were considered significant (hence low *p-values*).

## Computational modelling

Mathematical models were written to describe the mutually antagonistic effect of Ser46 and Ser88 phosphorylation on PKA and MAST3, respectively, as well as the direct inhibition from PKA to MAST3, and the 'dominant-negative' role of P-S88-ARPP-16 on PP2A inhibition. In these models, upon phosphorylation at Ser46 by MAST3, ARPP-16 becomes a stoichiometric inhibitor with high affinity binding, as well as being a substrate of PP2A. This results in low catalytic efficiency of PP2A. We hypothesized that P-S46-ARPP-16 inhibits PKA activity and lowers PKA catalytic efficiency, whereas P-S88-ARPP-16 inhibits MAST3 and weakens its catalytic efficiency as well. Our preliminary

experimental results indicate that phospho-Ser88 is not dephosphorylated by PP2A, and for the model we assumed that dephosphorylation at Ser88 was catalyzed by PP1. For modeling the direct inhibition from PKA to MAST3, we assumed that PKA not only inactivates MAST3, but inactivated MAST3 also interferes with active MAST3 phosphorylation of ARPP-16. Finally, we hypothesized that P-S88-ARPP-16 antagonizes PP2A inhibition by weakening the binding between P-S46-ARPP-16 and PP2A.

All phosphorylation and dephosphorylation reactions were modelled following Michaelis-Menten kinetics (see additional details in Appendix 1). The activation of PKA followed the Hill equation and the parameters were validated against published experimental data (*Zawadzki and Taylor, 2004*) (see *Appendix 1—figure 7*). Other regulations were modelled following laws of mass action. Inhibition of PP2A by P-S46-ARPP-16 and dephosphorylation of P-S46-ARPP-16 was modelled as described (*Vinod and Novak, 2015*). Parameters for PP1 were as described (*Hayer and Bhalla, 2005*). The total concentrations of each protein were estimated to correspond to their relative expression levels in striatum and were calculated relative to DARPP-32 abundance based on a recent mouse brain proteomic study (*Sharma et al., 2015*) (see *Appendix 1—table 2*). We derived the values of the kinetic constant Km for Ser46 and Ser88 phosphorylation based on double reciprocal plots of data from *Figure 1b and d*. Kinetic constants ($kcat_{PKA}$ and $kcat_{MAST3}$) and inhibitor constants (k88, k46, a and b) were estimated using the Particle Swam method implemented in the software COPASI (*Hoops et al., 2006*) and based on the data presented in *Figure 1a-d* (see Appendix 1—the mutual inhibition model and *Table 1*). Parameters for PKA inactivation of MAST3 ($k_{PKA}$) and how inactivated MAST3 interferes with catalytic efficiency of active MAST3 (r) were estimated as above, based on data presented in *Figure 4b* (see Appendix 1—the mutual inhibition plus PKA inhibits MAST3 model and *Table 1*). The parameter representing how P-S88-ARPP-16 antagonizing PP2A binding to P-S46-ARPP-16 (v) was estimated and validated by comparing simulation results with experimental data (see Appendix 1—the mutual inhibition plus PKA inibits MAST3 and dominant negative model and *Table 1*). Parameter estimation was performed using the SBPIPE package (*Dalle Pezze and Le Novère, 2017*). The optimum estimation results from five hundred trials were displayed for every possible pair of parameters under the 95% confidence interval of the best values (see Appendix 1—the first two models). The local minima reached in these estimations indicate that these parameters are identifiable for the given experimental data. Model equations and parameters are listed in Appendix 1. Bifurcation analysis was conducted with XPP-Aut (*Ermentrout, 2002*). The models are available in the BioModels Database (*Juty et al., 2015*)(MODEL1707020000, MODEL1707020001, MODEL1707020002).

## Acknowledgements

We would like to thank Mary LoPresti, Edward Voss, and Kathrin Wilczak for their assistance in MS sample preparation, and Piero Dalle Pezze for help with identifiability analysis and parameter estimation. Funding: This work was supported by NIH (DA10044 to ACN and PG). Proteomic analysis was supported by the Yale/NIDA Neuroproteomics Center (DA018343). VM was supported by a NAR-SAD Young Investigator award and NS091336. Support was also obtained from the State of Connecticut, Department of Mental Health and Addiction Services. LL was supported by the European Commission (grant 305299 AgedBrainSYSBIO). NL was supported by the Biotechnology and Biological Sciences Research Council (grant BBS/E/B/000C0419)

## Additional information

### Funding

| Funder | Grant reference number | Author |
| --- | --- | --- |
| National Institute on Drug Abuse | DA018343 | Angus C Nairn |
| National Institutes of Health | DA10044 NS091336 | Veronica Musante Paul Greengard Angus C Nairn |
| European Commission | 305299 | Lu Li |

| State of Connecticut Department of Mental Health and Addiction Services | | Angus C Nairn |
|---|---|---|
| Biotechnology and Biological Sciences Research Council | BBS/E/B/000C0419 | Nicolas Le Novère |
| Brain and Behavior Research Foundation | | Veronica Musante |

The funders had no role in study design, data collection and interpretation, or the decision to submit the work for publication.

### Author contributions

VM, Conceptualization, Data curation, Formal analysis, Funding acquisition, Investigation, Writing—original draft, Project administration, Writing—review and editing; LL, Computational modelling, Data curation, Formal analysis, Visualization, Methodology, Writing—original draft; JK, CMC, Formal analysis, Methodology; TTL, Formal analysis, Methodology, Writing—original draft; SKC, AHB, Methodology; PG, Resources; NLN, Computational modelling, Conceptualization, Data curation, Writing—original draft; ACN, Conceptualization, Supervision, Funding acquisition, Writing—original draft, Project administration, Writing—review and editing

### Author ORCIDs

Veronica Musante, http://orcid.org/0000-0001-5244-9158
Lu Li, http://orcid.org/0000-0002-6199-522X
Nicolas Le Novère, http://orcid.org/0000-0002-6309-7327
Angus C Nairn, http://orcid.org/0000-0002-7075-0195

## Additional files

### Supplementary files

• Supplementary file 1. Summary of phospho-peptides identified by LC-MS/MS: The sequences of phospho-peptides identified in HA-MAST3 isolated from HEK293T cell experiments are listed in order of amino acid residue position in the protein sequence. Data from six different experiments (labelled as 1–6) are from the various digestion conditions utilized as described in the Materials and methods section. Samples from HEK293T cells incubated under control conditions (CTRL) or in the presence of forskolin (FSK) are shown. Criteria for positive phospho site identification (Primary or Alternative) includes (1) meeting PhosphoRS criteria; (2) Mascot expectation score of 0.05 or less; and peptide mass accuracy of 10 ppm or less.

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

## Appendix 1

## Description of the mathematical models

### Mutual inhibition between P-S46-ARPP-16 and P-S88-ARPP-16

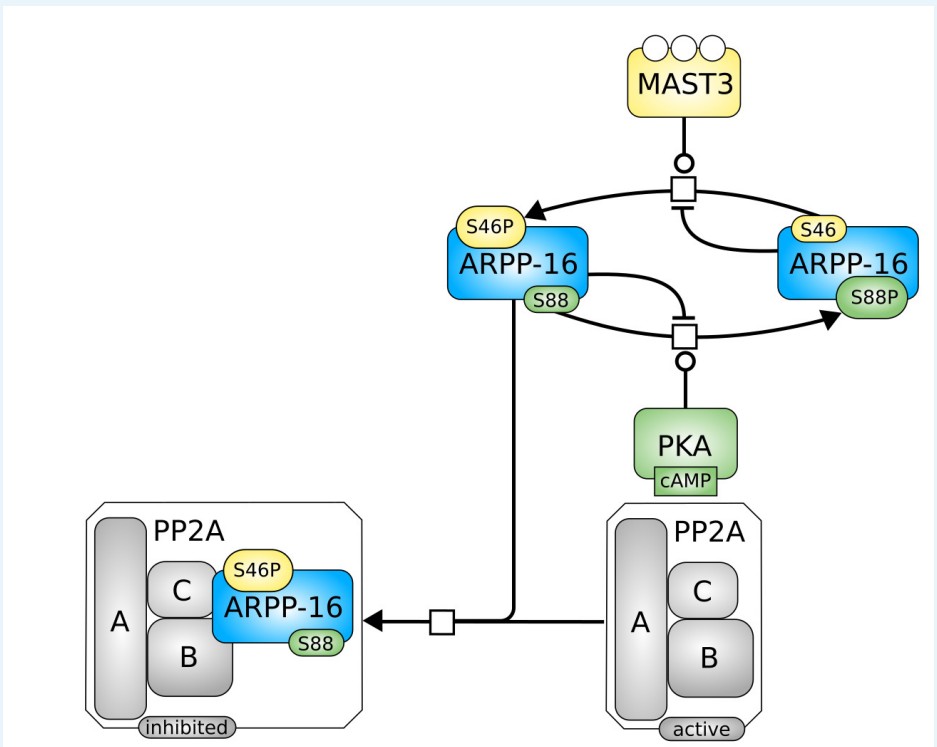

**Appendix 1—scheme 1.** SBGN schema of the model including the mutual inhibition between P-S46-ARPP-16 and P-S88-ARPP-16, was well as the direct action of PKA on ARPP-16.

### Equations

$$Km \approx Kd = \frac{(A46 - A46 : PP2A) \cdot (PP2A_{tot} - A46 : PP2A)}{A46 : PP2A}$$

$$A46 : PP2A = \frac{1}{2} \cdot \left( A46 + PP2A_{tot} + Km - \sqrt{(A46 + PP2A_{tot} + Km)^2 - 4 \cdot A46 \cdot PP2A_{tot}} \right)$$

$$\frac{dA46}{dt} = \frac{kcat_{MAST3} \cdot MAST3 \cdot (A_{tot} - A46)}{Km_{MAST3} + a \cdot \frac{A88}{A_{tot}} + (A_{tot} - A46)} - kcat_{PP2A} \cdot A46 : PP2A$$

$$\frac{dA88}{dt} = \frac{kcat_{PKA} \cdot PKA \cdot (A_{tot} - A88)}{Km_{PKA} + b \cdot \frac{A46}{A_{tot}} + (A_{tot} - A88)} - \frac{kcat_{PP1} \cdot PP1 \cdot A88}{Km_{PP1} + A88}$$

$$\frac{\text{dMAST3}}{dt} = k_{\text{PPX}} \cdot (\text{MAST3}_{\text{tot}} - \text{MAST3}) - k88 \cdot \text{A88} \cdot \text{MAST3}$$

$$\frac{d\text{PKA}}{dt} = \frac{k_{\text{cAMP}} \cdot (\text{PKA}_{\text{tot}} - \text{PKA}) \cdot \text{cAMP}^n}{KA^n + \text{cAMP}^n} - k46 \cdot \text{A46} \cdot \text{PKA}$$

A46 concentration of P-S46-ARPP-16

A46:PP2A concentration of the complex between P-S46-ARPP-16 and PP2A

A88 concentration of P-S88-ARPP-16

$A_{\text{tot}}$ total concentration of ARPP-16

## Distributions of estimated parameters converge towards optimal values.

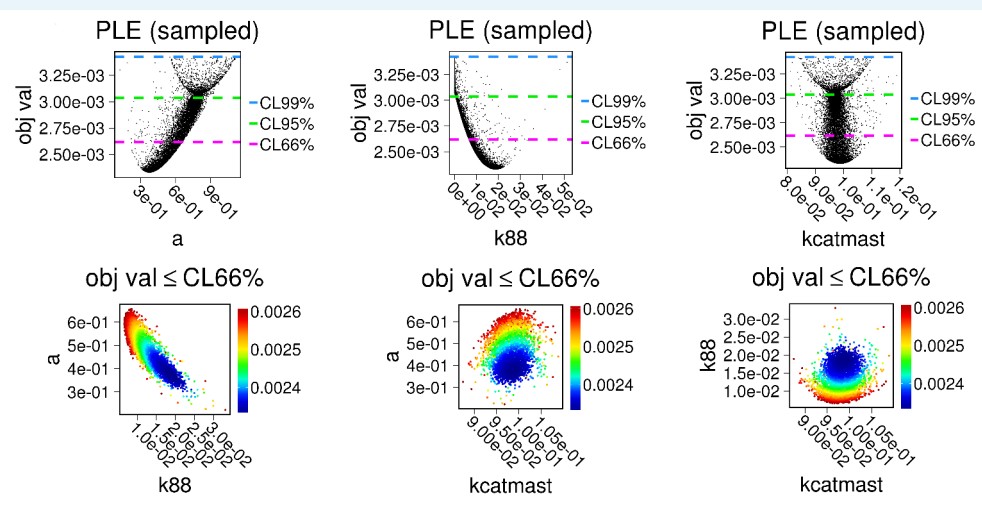

**Appendix 1—figure 1.** First row: parameter estimation results displayed against chi-square score; Second row: estimations for each pair of parameters showing identifiability. Parameters a, k88 and kcat$_{\text{mast}}$ were estimated using the Particle Swam method 500 times, based on data from main Figure 1a,b. To match with experimental conditions, concentrations of PKA, and all the phosphatases were set to be zero in the model.

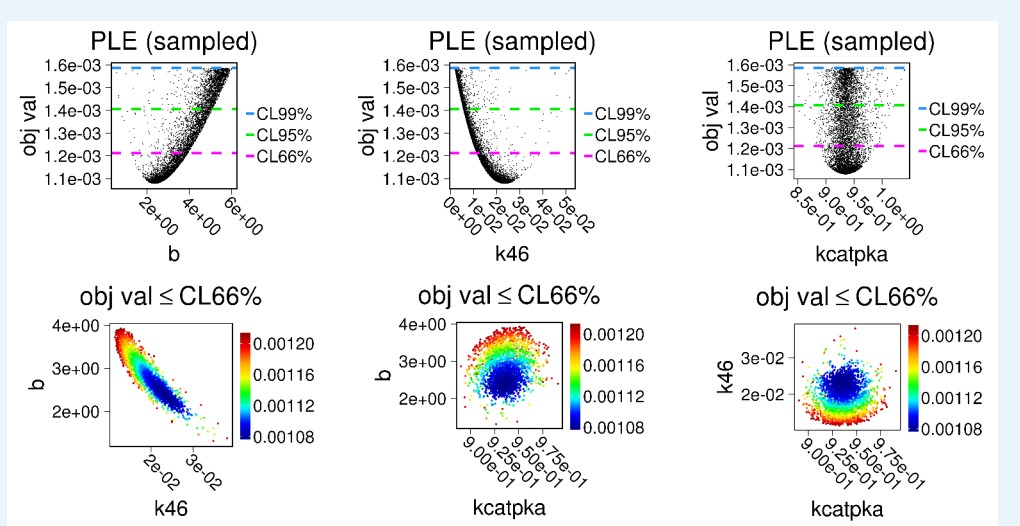

**Appendix 1—figure 2.** First row: parameter estimation results displayed against chi-square score; Second row: estimations for each pair of parameters showing identifiability. Parameters b, k46, kcat$_{PKA}$ were estimated as above based on data from main *Figure 1 c,d*. To match experimental conditions, concentrations of MAST3 and all phosphatases were set to be zero in the model.

Parameter estimation results.

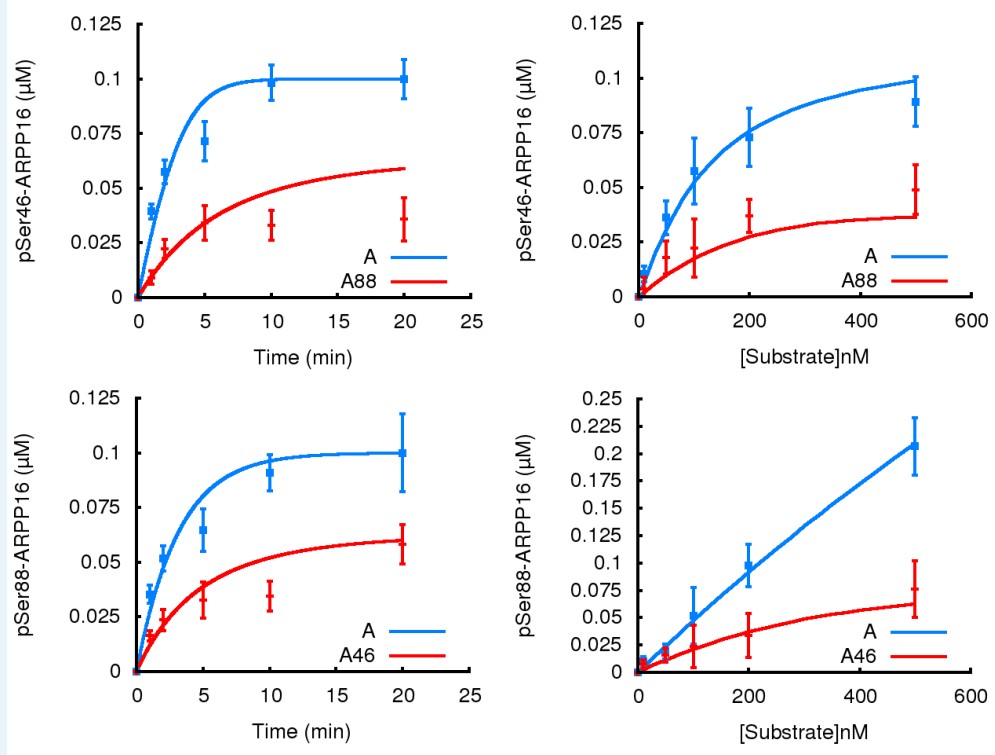

**Appendix 1—figure 3.** Model simulation results compared with experiments presented in main *Figure 1a–d* (mean±SD).

## Mutual inhibition + PKA's direct regulation on MAST3

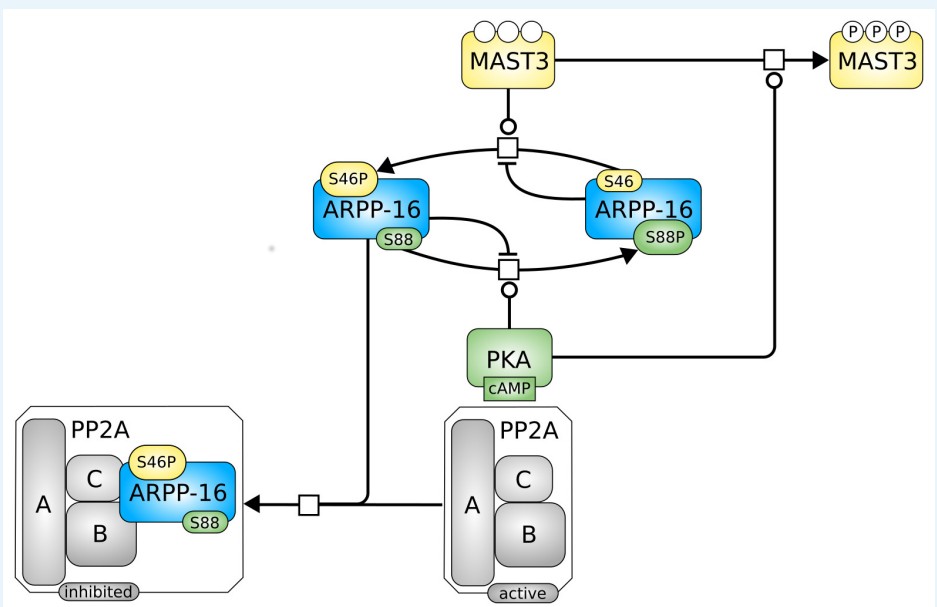

**Appendix 1—scheme 2.** SBGN schema of the model including the mutual inhibition between phosphorylated ARPP-16, the direct action of PKA on ARPP-16, as well as PKA phosphorylation of MAST3.

## Equations

$$Km \approx Kd = \frac{(\text{A46} - \text{A46}:\text{PP2A}) \cdot (\text{PP2A}_{\text{tot}} - \text{A46}:\text{PP2A})}{\text{A46}:\text{PP2A}}$$

$$\text{A46}:\text{PP2A} = \frac{1}{2}\left(\text{A46} + \text{PP2A}_{\text{tot}} + Km - \sqrt{(\text{A46} + \text{PP2A}_{\text{tot}} + Km)^2 - 4 \cdot \text{A46} \cdot \text{PP2A}_{\text{tot}}}\right)$$

$$\frac{d\text{A46}}{dt} = \frac{kcat_{\text{MAST3}} \cdot \text{MAST3} \cdot (\text{A}_{\text{tot}} - \text{A46})}{Km_{\text{MAST3}} + a \cdot \frac{\text{A88}}{\text{A}_{\text{tot}}} + r \cdot \frac{(\text{MAST3}_{\text{tot}} - \text{MAST3})}{\text{MAST3}_{\text{tot}}} + (\text{A}_{\text{tot}} - \text{A46})} - kcat_{\text{PP2A}} \cdot \text{A46}:\text{PP2A}$$

$$\frac{d\text{A88}}{dt} = \frac{kcat_{\text{PKA}} \cdot \text{PKA} \cdot (\text{A}_{\text{tot}} - \text{A88})}{Km_{\text{PKA}} + b \cdot \frac{\text{A46}}{\text{A}_{\text{tot}}} + (\text{A}_{\text{tot}} - \text{A88})} - \frac{kcat_{\text{PP1}} \cdot \text{PP1} \cdot \text{A88}}{Km_{\text{PP1}} + \text{A88}}$$

$$\frac{d\text{MAST3}}{dt} = k_{\text{PPX}} \cdot (\text{MAST3}_{\text{tot}} - \text{MAST3}) - k88 \cdot \text{A88} \cdot \text{MAST3} - k_{\text{PKA}} \cdot \text{PKA} \cdot \text{MAST3}$$

$$\frac{d\text{PKA}}{dt} = \frac{k_{\text{cAMP}} \cdot (\text{PKA}_{\text{tot}} - \text{PKA}) \cdot \text{cAMP}^n}{KA^n + \text{cAMP}^n} - k46 \cdot \text{A46} \cdot \text{PKA}$$

Distributions of estimated parameters converge at the best values.

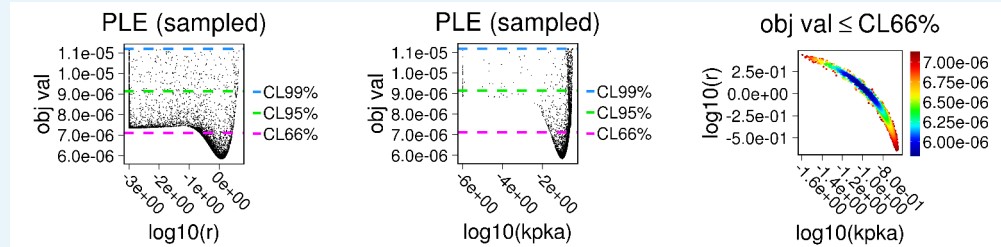

**Appendix 1—figure 4.** First two panels: parameter estimation results displayed against chi-square score; last panel: estimations for this pair of parameters showing identifiability. Parameters r and $k_{PKA}$ were estimated following as above based on data from main *Figure 4b*. To match with experimental conditions, concentrations of all phosphatases were set to zero in the model.

Parameter estimation results.

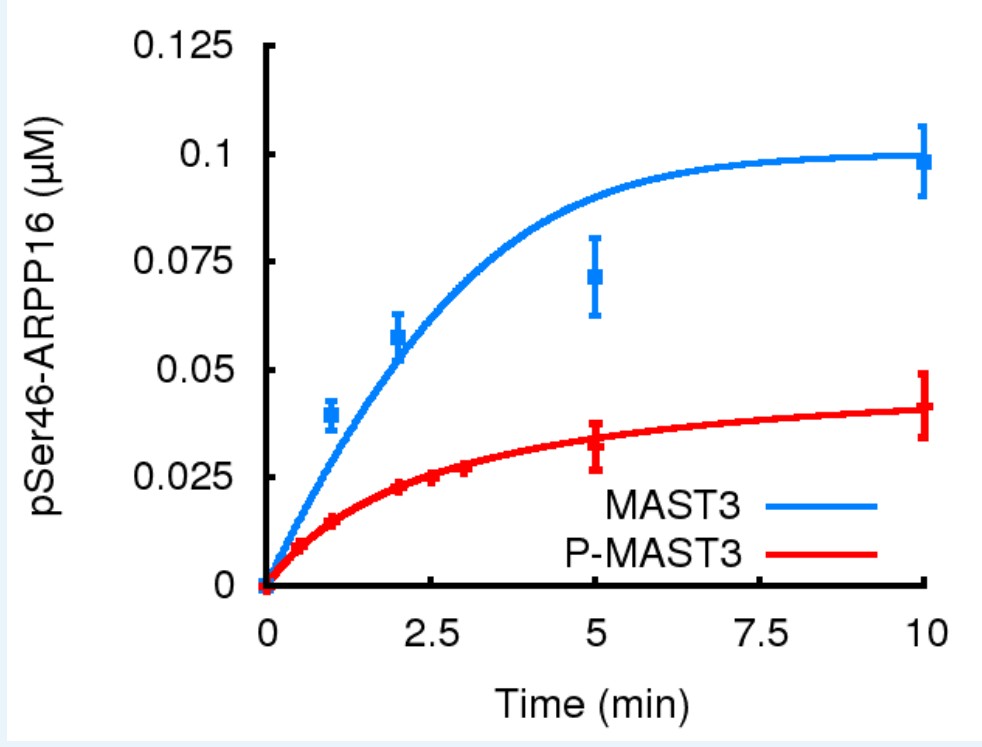

**Appendix 1—figure 5.** Model simulation results compared with experiments presented on *Figure 4b* (mean±SD).

## Mutual inhibition + PKA inhibits MAST3 + P-S88-ARPP-16 dominant negative effect

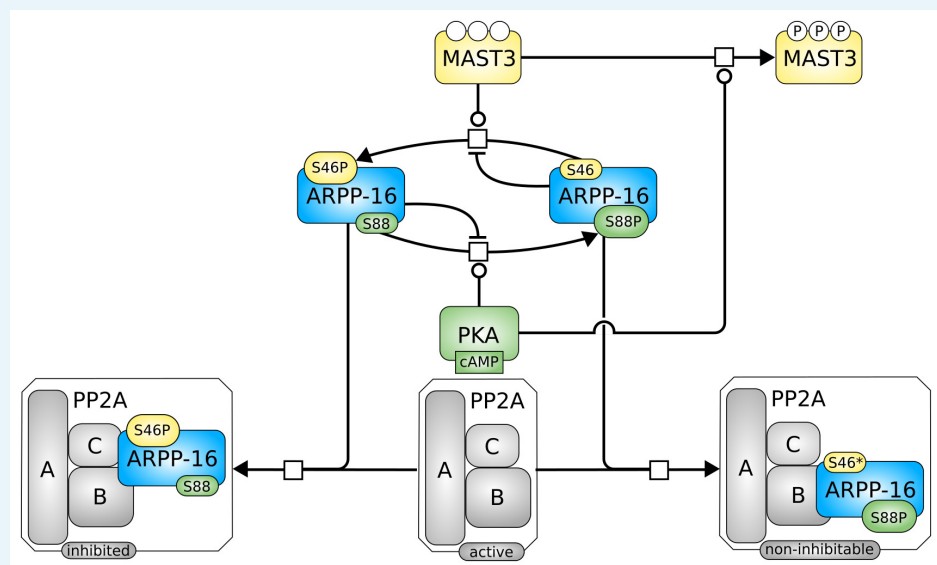

**Appendix 1—scheme 3.** SBGN schema of the model including the mutual inhibition between phosphorylated ARPP-16, the direct action of PKA on ARPP-16, PKA phosphorylation of MAST3, and the dominant negative effect of P-S88-ARPP-16.

## Equations

When only mutual inhibition and P-S88-ARPP-16 dominant negative effect were considered, the red parts of the equations were not included.

$$Km \approx Kd = \frac{(\mathrm{A46} - \mathrm{A46:PP2A}) \cdot (\mathrm{PP2A_{tot}} - \mathrm{A46:PP2A})}{\mathrm{A46:PP2A}}$$

$$
\begin{aligned}
&\mathrm{A46:PP2A} \\
&= \frac{\mathrm{A46} + \mathrm{PP2A_{tot}} + Km\left(1 + v\frac{\mathrm{A88}}{\mathrm{A_{tot}}}\right) - \sqrt{\left(\mathrm{A46} + \mathrm{PP2A_{tot}} + Km\left(1 + v\frac{\mathrm{A88}}{\mathrm{A_{tot}}}\right)\right)^2 - 4 \cdot \mathrm{A46} \cdot \mathrm{PP2A_{tot}}}}{2}
\end{aligned}
$$

$$\frac{d\mathrm{A46}}{dt} = \frac{kcat_{\mathrm{MAST3}} \cdot \mathrm{MAST3} \cdot (\mathrm{A_{tot}} - \mathrm{A46})}{Km_{\mathrm{MAST3}} + a \cdot \frac{\mathrm{A88}}{\mathrm{A_{tot}}} + r \cdot \frac{(\mathrm{MAST3_{tot}} - \mathrm{MAST3})}{\mathrm{MAST3_{tot}}} + (\mathrm{A_{tot}} - \mathrm{A46})} - kcat_{\mathrm{PP2A}} \cdot \mathrm{A46:PP2A}$$

$$\frac{d\mathrm{A88}}{dt} = \frac{kcat_{\mathrm{PKA}} \cdot \mathrm{PKA} \cdot (\mathrm{A_{tot}} - \mathrm{A88})}{Km_{\mathrm{PKA}} + b \cdot \frac{\mathrm{A46}}{\mathrm{A_{tot}}} + (\mathrm{A_{tot}} - \mathrm{A88})} - \frac{kcat_{\mathrm{PP1}} \cdot \mathrm{PP1} \cdot \mathrm{A88}}{Km_{\mathrm{PP1}} + \mathrm{A88}}$$

$$\frac{d\mathrm{MAST3}}{dt} = k_{\mathrm{PPX}} \cdot (\mathrm{MAST3_{tot}} - \mathrm{MAST3}) - k88 \cdot \mathrm{A88} \cdot \mathrm{MAST3} - k_{\mathrm{PKA}} \cdot \mathrm{PKA} \cdot \mathrm{MAST3}$$

$$\frac{d\mathrm{PKA}}{dt} = \frac{k_{\mathrm{cAMP}} \cdot (\mathrm{PKA}_{tot} - \mathrm{PKA}) \cdot \mathrm{cAMP}^n}{KA^n + \mathrm{cAMP}^n} - k46 \cdot \mathrm{A46} \cdot \mathrm{PKA}$$

Parameter estimation results.

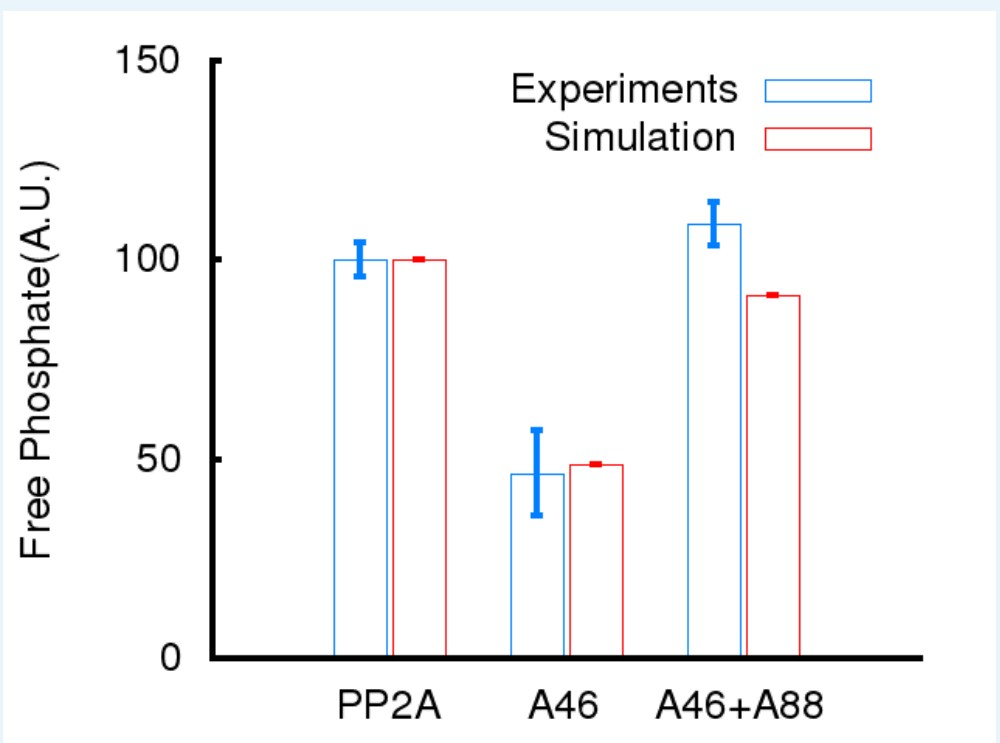

**Appendix 1—figure 6.** Simulation results compared with experiments presented in main *Figure 7*.

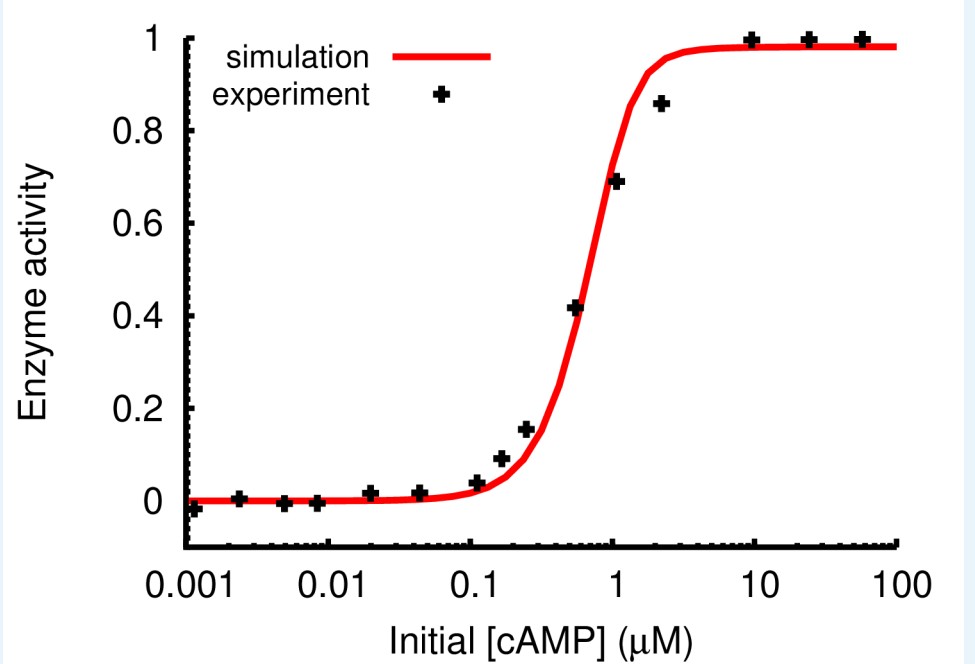

**Appendix 1—figure 7.** Parameters used in the Hill-equation encoding PKA activation by cAMP were validated against experimental observations (*Zawadzki and Taylor, 2004*). Simulation results were obtained by mixing 10 nM of PKA whole enzyme with 200 µM kemptide and

cAMP of varying concentrations as described in the published experiments. The enzyme activity was shown as the normalized level of phospho-kemptide after a 2 min simulation.

**Appendix 1—table 1.** Parameter values.

| Equation | Parameter | Value | Reference |
|---|---|---|---|
| 1 | $K_d$ | 1 nM | *Vinod and Novak (2015)* |
| | v | 100 | Estimated and validated using data from *Figure 7* |
| 2 | $kcat_{MAST3}$ | $0.0988\ s^{-1}$ | Estimated using data from *Figure 1a,b* |
| | $Km_{MAST3}$ | 0.09 μM | Obtained from *Figure 1b* |
| | a | 0.37526 | Estimated using data from *Figure 1a,b* |
| | r | 1.2 | Estimated using data from *Figure 4b* |
| | $kcat_{PP2A}$ | $0.05\ s^{-1}$ | *Vinod and Novak (2015)* |
| 3 | $kcat_{PKA}$ | $0.935\ s^{-1}$ | Estimated using data from *Figure 1c,d* |
| | $Km_{PKA}$ | 1.6 μM | Obtained from *Figure 1d* |
| | b | 2.36 | Estimated using data from *Figure 1c,d* |
| | $kcat_{PP1}$ | $0.5\ s^{-1}$ | Estimated using data from *Hayer and Bhalla (2005)* |
| | $Km_{PP1}$ | 1 μM | Estimated using data from *Hayer and Bhalla (2005)* |
| 4 | $k_{ppx}$ | $0.05\ s^{-1}$ | Estimated in this study |
| | k88 | $0.01865\ \mu M^{-1}s^{-1}$ | Estimated using data from *Figure 1a,b* |
| | $k_{PKA}$ | 0.097 | Estimated using data from *Figure 4b* |
| 5 | $k_{cAMP}$ | $0.7\ s^{-1}$ | Estimated and validated by *Zawadzki and Taylor (2004)* |
| | n | 2 | Estimated and validated by *Zawadzki and Taylor (2004)* |
| | KA | 10 μM | Estimated and validated by *Zawadzki and Taylor (2004)* |
| | k46 | $0.02335\ \mu M^{-1}.s^{-1}$ | Estimated using data from *Figure 1c,d* |

**Appendix 1—table 2.** Initial concentrations.

| **PP2Atot** | **2 μM** |
|---|---|
| Atot | 10 μM |
| MAST3tot | 2.7 μM |
| PKAtot (total catalytic subunits) | 12 μM |
| PP1 | 5 μM |

