## [Decision Letter]

Thank you for submitting your article "Regulation of ARPP-16 by PKA and MAST3 kinases provides a cAMP-regulated switch in phosphatase PP2A inhibition" for consideration by *eLife*. Your article has been reviewed by three peer reviewers, one of whom is a member of our Board of Reviewing Editors, and the evaluation has been overseen by Jonathan Cooper as the Senior Editor. The following individual involved in review of your submission has agreed to reveal his identity: John D Scott (Reviewer #2).

The reviewers have discussed the reviews with one another and the Reviewing Editor has drafted this decision to help you prepare a revised submission..

Summary:

This is an interesting manuscript that describes a study that was designed to examine the interplay between MAST3 and PKA signaling in the regulation of PP2A by ARRP-16.

Previous studies have established that ARPP-19 is phosphorylated and inhibited by PKA (Dupre et al., 2014) and that PKA phosphorylation of ARPP-16/19 on Ser88 is associated with dephosphorylation of ARPP-16/19 phosphorylation on the MAST3 site Ser46 that mediates PP2A inhibition (Andrade et al., in revision).

The present study seeks to identify mechanisms that mediate the inhibitory actions of PKA on MAST3-regulated PP2A activity. Three different mechanisms are described. First, ARPP-16 phosphorylated more slowly by PKA and MAST3, if either Ser46 or Ser88, respectively, is already phosphorylated. This is an elegant mechanism of cross-regulation, presumably induced by changes in substrate conformation. Second, PKA phosphorylates and inhibits MAST3. Third, phosphorylation of ARPP-16 at Ser88 can inhibit the effects of phosphoSer46 ARPP-16 on PP2A inhibition by a dominant-negative mechanism in trans. Collectively, these data indicate that there are multiple levels of cross-regulation of MAST3 signaling to ARPP-16 by PKA.

Essential revisions:

1) While three different forms of cross-regulation by PKA are identified, this is not apparent in the abstract of the manuscript. Moreover, the computational model presented does not capture this complexity. Furthermore, this complexity is not illustrated in the final cartoon model presented in Figure 8.

2) The Supplementary Materials support the mathematical model written and employed, with respect to adequate fitting of time-course data and cAMP-dependence of PKA activity. The model itself appears to be satisfactory under various assumptions essentially setting other cell biochemistry processes irrelevant. The steady-state bifurcation analysis described is appropriate. However, as noted above, the model does not take account of MAST3 inhibition by PKA or the dominant-negative effects of ARPP-16 phosphorylation. The importance of the model comes from predictions derived from the model that can be experimentally tested. For example, what happens if one tests the hypothesis upon stimulation with isoproterenol, dopamine or epinephrine?. Or inhibits PKA with H89 or PKI peptides? Also, are the results skewed upon deletion or silencing MAST3. This latter experiment is important to establish if other kinases compliment for MAST3 activity inside cells.

3) The biochemical studies in Figure 1 are well performed and provide the first evidence for the model proposed by the authors. However, the next phase of the analyses uses overexpression of proteins in HEK293 cells. While this is a slightly more physiologically significant system it would be prudent to include data showing the phenomena in primary cultures of neurons.

4) Another weakness of the experiments in Figure 2 is that forskolin (a direct activator of Adenylyl cyclases) is used as a means to over stimulate cAMP synthesis. This creates a situation where cells accumulate cAMP at levels that are 15-20 fold above the normal physiological range. Can experiments be done in HEK293 cells with more natural agonists such as isoproterenol, dopamine or epinephrine.

5) The cross-inhibitory effects of Ser46 and Ser88 phosphorylation on subsequent ARPP-16 phosphorylation and PP2A inhibition are interesting. These observations raise questions concerning whether dual phosphorylated endogenous ARPP-16 is found in vivo. This could be tested by immunoprecipitation/western blot analysis using phospho-specific antibodies.

6) PKA phosphorylates and inhibits MAST3 in in vitro (Figure 4). Does cAMP signaling in vivo inhibit endogenous MAST3 activity?

7) PKA appears to inhibit MAST3 activity in HEK transfection assays using ARPP-16 Ser46 as a read-out (Figure 5 and Figure 6). Studies using phosphorylation site mutations are unclear. Presumably, if the phosphorylation sites were important for MAST3 activity, there should be major differences MAST3 activity between the WT and Ser/Ala mutants in the presence of activated PKA signaling. However, the data presented shows that the WT and mutated MAST3 proteins cause similar Ser46 ARPP-16 phosphorylation in each case. Please clarify.

8) The ability of Ser88 phosphorylated ARPP-16 to act as a dominant inhibitor of Ser64 phosphorylated ARPP-16 is intriguing. This suggests that the two ARPP-16 proteins might compete for binding to PP2A and that pSer88 ARP-16 has a higher affinity for PP2A than pSer64 ARPP-16. Alternatively, the binding may be non-competitive and pSer88 ARPP-16 may have dominant activity. Other mechanisms are also possible. Studies to evaluate mechanism are required, including the determination of binding affinity and binding competition.

---

## [Author Response]

*Essential revisions:*

*1) While three different forms of cross-regulation by PKA are identified, this is not apparent in the abstract of the manuscript. Moreover, the computational model presented does not capture this complexity. Furthermore, this complexity is not illustrated in the final cartoon model presented in Figure 8.*

Abstract – The Abstract has been edited to describe more fully the content of the manuscript. Note that due to the 150-word limit numerous edits/deletions were made to the text.

Computational model – As described in the response to comment #2, we have made significant additions to the computational modeling as suggested by the reviewers.

Figure 8 – The three different forms of cross-regulation of PP2A by PKA were included in the original Figure 8, but we did not explain this clearly in the legend. The legend has been expanded, and we have also revised the text in the Discussion related to Figure 8.

*2) The Supplementary Materials support the mathematical model written and employed, with respect to adequate fitting of time-course data and cAMP-dependence of PKA activity. The model itself appears to be satisfactory under various assumptions essentially setting other cell biochemistry processes irrelevant. The steady-state bifurcation analysis described is appropriate. However, as noted above, the model does not take account of MAST3 inhibition by PKA or the dominant-negative effects of ARPP-16 phosphorylation.*

We modified and re-estimated the parameters in the original mathematical model representing the reciprocal inhibitions (layer 1), using not only the time-course experimental data (Figure 1), but also the dose-response experiments (Figure 1). These changes provide a better match between simulation results from the model and experimental observations, while keeping intact the switch response to cAMP at similar dynamic cAMP concentration ranges (Appendix Section).

We then extended the model to include MAST3 inhibition by PKA (layer 2) and the dominant negative effect of P-S88-ARPP-16 on PP2A inhibition (layer 3). We estimated the corresponding parameters against our own experiments (Appendix Section). Model simulations showed that these additional layers of PKA regulation maintain bistability and switch-like control over PP2A activity. However these regulations shift the cAMP dynamic range for bistability to a more sensitive micro-molar level, and most importantly, deepen the PP2A dis-inhibition when cAMP increases above the threshold (see Figure 3—figure supplement 1 and Figure 3—figure supplement 2).

We further studied how total MAST3 concentration change affects switch-like control of PP2A activity, as MAST3 or other MAST isoforms are expressed at different levels in different brain regions and potentially could exert similar effects on PP2A. Our simulation results revealed that there is a wide range of total MAST3 concentration that supports bistability and therefore switch-like control over PP2A inhibition (see Figure 3—figure supplement 3, Figure 3—figure supplement 4, Figure 3—figure supplement 5).

Finally we briefly discuss the potential impact of the striatal PP1 inhibitor, DARPP-32, that can potentially modify the cAMP dynamic concentration range through its phosphorylation at Thr75 and inhibition of PKA. The antagonistic inhibition from P-S46-ARPP-16 to Ser88 phosphorylation can be strengthened by the negative regulation from Ser46 phosphorylation to PKA activity, via PP2A and Thr75-DARPP-32. Our simulations showed that the bistability and switch like control over PP2A is robust (Figure 3—figure supplement 6 and Figure 3—figure supplement 7). However the dynamic range of cAMP concentration controlling the on and off PP2A inhibition is potentially wider (more hysteresis) than what we revealed here.

*The importance of the model comes from predictions derived from the model that can be experimentally tested. For example, what happens if one tests the hypothesis upon stimulation with isoproterenol, dopamine or epinephrine?. Or inhibits PKA with H89 or PKI peptides? Also, are the results skewed upon deletion or silencing MAST3. This latter experiment is important to establish if other kinases compliment for MAST3 activity inside cells.*

This part of Comment #2 overlaps a bit with Comments #3 and #4, and we apologize for any repetition. We fully agree that one utility of the modeling is to test predictions drawn from the model in an experimental system. However, there are practical limits to the experimental systems we have available. Although we did not add detailed information to the original text, we find that the expression levels of the components of the ARPP/MAST3/PKA/PP2A signaling system are high in adult striatum, but the key proteins, ARPP-16/19/ENSA and MAST3 are very low in cultured striatal neurons. ARPP-19 is expressed in Hek cells to ~50% of the level of ARPP-16/19/ENSA in adult striatum (Figure 19) but MAST3 is undetectable in Hek cells (Figure 19). Trying to manipulate expression of multiple proteins in primary cultures would be challenging. We therefore chose to establish conditions of ARPP-16/19 and MAST3 expression in Hek cells to be comparable to that seen in adult striatum (Figure 19). This puts constraints on potential experimental manipulations in Hek cells, with exposure to forskolin being the primary variable used to stimulate cAMP from low basal to high experimental levels. The only other alternative approach available is to use acutely prepared striatal slices from adult rodent, an ex vivo model. These have the intrinsic advantage of being a more physiological representation of adult striatum (see below) but are only amenable to pharmacological manipulation.

Author response image 1.The expression levels of endogenous ARPP-16 (panel a.) and MAST3 (panel b.) in striatum were compared to ARPP16-HA, MAST3-HA (a shorter form of MAST3 used in the studies described in the manuscript) and full length MAST3-FL-HA recombinant proteins expressed in HEK293 cell.Protein expression was measured by serial immunoblotting using an antibody against MAST3 (to detect endogenous protein in striatum and cells overexpressing MAST3-FL-HA) and an antibody against the HA tag (to compare the levels of MAST3-HA and MAST3-FL-HA). Immunoblots were carried out in a similar range of protein amounts and protein levels were normalized to GAPDH.**DOI:**
http://dx.doi.org/10.7554/eLife.24998.033

Stimulation with isoproterenol, dopamine or epinephrine – Prior studies have indicated that isoproterenol or epinephrine could be used to increase cAMP levels in Hek cells [e.g. (Schmitt and Stork, 2000; Friedman et al., 2002; Shenoy et al., 2006)], but these studies typically require long-term serum deprivation, pretreatment with pertussis toxin, or over-expression of GPCRs. Under the conditions used for our Hek cell studies, we incubated cells with isoproterenol but we could not see any consistent stimulation of phosphorylation of ARPP-16 by PKA. As an alternative, we used striatal slices and showed that ARPP-16 and ARPP-19 were phosphorylated at high basal levels by MAST3. Stimulation with SKF81297 (a D1 receptor agonist), like forskolin, led to rapid increase in S88-ARPP-16 phosphorylation, which was accompanied by decreased phosphorylation of S46. We have added this data to the manuscript as Figure 2—figure supplement 1.

Inhibition of PKA with H89 or PKI peptides – In striatal slices and the Hek cell model, cAMP is basally low, and as a result S88 is low. Inhibition of PKA would not be expected to have any effect.

Deletion or silencing MAST3 – Given the experimental systems available, we were limited on options (within a reasonable time frame) for deletion or silencing MAST3 in striatal neurons, which would require generating a (preferably) cell type-specific striatal knockout mouse or using viral-mediated manipulation in mouse striatum. We were recently aware of structural work on Greatwall kinase, in which a small molecule inhibitor was generated that appeared to inhibit MAST-L (Ocasio et al., 2016). As an alternative approach to address this comment, we obtained some of the Greatwall inhibitor from these investigators, but it had no effect on MAST3 activity in vitro or on S46 phosphorylation in Hek cells or striatal slices. Our future studies will include targeted silencing of MAST-3 in medium spiny neurons, coupled with biochemical and behavioral studies. As noted above, we did however carry out additional modeling analysis, and found that, over a 4-fold range, varying total MAST3 concentration did not affect the general switch-like response to cAMP concentration changes (Figure 3—figure supplement 3, 4 and 5). This information has been added to the Discussion section.

*3) The biochemical studies in Figure 1 are well performed and provide the first evidence for the model proposed by the authors. However, the next phase of the analyses uses overexpression of proteins in HEK293 cells. While this is a slightly more physiologically significant system it would be prudent to include data showing the phenomena in primary cultures of neurons.*

As noted above, we examined the levels of the key proteins, ARPP-16/19 and MAST3 in primary cultures that are enriched for striatal neurons, but there was no detectable expression of ARPP-16 and only low levels of expression of ENSA compared to that found in adult striatum. Similarly, the expression of MAST3 was very low in cultured striatal neurons compared to adult striatum. The level of ARPP-16 expression is known to be highly regulated during postnatal brain development, with adult brain levels being reached after several postnatal weeks (Girault et al., 1990). As a result, we decided to use a Hek cell system, where we established conditions to be similar to that seen in adult striatum in terms of ARPP-16/19 and MAST3 expression, with the corresponding basal levels of high S46 phosphorylation and low S88 phosphorylation (Figure 19 and figures in manuscript). As also noted above, we acutely prepared striatal slices and carried out some studies that involve pharmacological manipulation. ARPP-16 and ARPP-19 were phosphorylated at high basal levels by MAST3. Stimulation with SKF81297 (a D1 receptor agonist), like forskolin, led to an increase in S88-ARPP-16 phosphorylation, which was accompanied by decreased phosphorylation of S46. We have added this data to the manuscript as Figure 2—figure supplement 1, as part of a reorganized introduction to the second part of the Results section, where we discuss our rationale for using Hek cells.

*4) Another weakness of the experiments in Figure 2 is that forskolin (a direct activator of Adenylyl cyclases) is used as a means to over stimulate cAMP synthesis. This creates a situation where cells accumulate cAMP at levels that are 15-20 fold above the normal physiological range. Can experiments be done in HEK293 cells with more natural agonists such as isoproterenol, dopamine or epinephrine.*

We recognize the limitations of using forskolin. As noted above, we attempted to use isoprotereonol in Hek cells but were unable to detect reliable increased phosphorylation of S88-ARPP-16. We think this is a result of low expression of the requisite GPCR [see for example Figure 6 in Shenoy et al., 2006]. We considered trying to overexpress a GPCR as done in other studies, but thought that this would not really help address the reviewer’s comment. As an alternative, as noted above, we used striatal slices and showed that stimulation with SKF81297 (a D1 receptor agonist) led to a rapid increase in S88-ARPP-16 phosphorylation, which was accompanied by decreased phosphorylation of S46. This effect was similar to that of forskolin. We have added this data to the manuscript as Figure 2—figure supplement 1.

*5) The cross-inhibitory effects of Ser46 and Ser88 phosphorylation on subsequent ARPP-16 phosphorylation and PP2A inhibition are interesting. These observations raise questions concerning whether dual phosphorylated endogenous ARPP-16 is found in vivo. This could be tested by immunoprecipitation/western blot analysis using phospho-specific antibodies.*

This is an interesting question and one that we had considered. As suggested by the reviewer, we attempted to use the phospho-S46 and phospho-S88 antibodies for immunoprecipitation, but unfortunately this was not successful. The two antibodies used are affinity purified, polyclonal rabbit antibodies. While motif-based phospho-serine/threonine antibodies have been used for enrichment of short phosphopeptides, and phospho-tyrosine antibodies are commonly used for immunoprecipitation, we do not know why our antibodies do not immunoprecipitate our holo-proteins. Presumably, the solution-based conditions used for immunoprecipitation are sufficiently different than that used for affinity purification with short epitopes, or for immunoblotting.

*6) PKA phosphorylates and inhibits MAST3 in in vitro (Figure 4). Does cAMP signaling in vivo inhibit endogenous MAST3 activity?*

Such in vivo studies would require using rodents, and assessing S46-ARPP-16 phosphorylation in striatum as a proxy for MAST3 activity. We have carried out preliminary studies of this type using the psychostimulant, cocaine, to stimulate dopamine receptors and cAMP levels in striatum. The preliminary results showed that acute exposure to cocaine reduced pS46. As an alternative, as noted above, we use here an ex vivo preparation, namely striatal slices, to show that stimulation with SKF81297 (a D1 receptor agonist) led to a rapid decrease in phosphorylation of S46. We have added this data to the manuscript as Figure 2—figure supplement 1.

*7) PKA appears to inhibit MAST3 activity in HEK transfection assays using ARPP-16 Ser46 as a read-out (Figure 5 and Figure 6). Studies using phosphorylation site mutations are unclear. Presumably, if the phosphorylation sites were important for MAST3 activity, there should be major differences MAST3 activity between the WT and Ser/Ala mutants in the presence of activated PKA signaling. However, the data presented shows that the WT and mutated MAST3 proteins cause similar Ser46 ARPP-16 phosphorylation in each case. Please clarify.*

We apologize for not explaining the results clearly, and we have reorganized and edited this section of the Results. Our initial studies indicated that PKA phosphorylates MAST3 at multiple sites, both in vitro and in intact cells. In vitro, PKA-dependent phosphorylation of MAST3 is associated with inhibition. We then investigated the roles of specific phosphorylation sites identified by mass spectrometry and bioinformatics (originally shown in Figure 5, Figure 6 and Figure 3—figure supplement 5). To assess the effect of one of the sites, Thr389, we immunoprecipitated WT, T389D-MAST3 and T389A-MAST3 from Hek cells, and carried out in vitro assays to directly assess the effects of PKA-dependent phosphorylation on MAST3 activity. The T389D-MAST3-HA mutant was much less active than WT-MAST3-HA (Figure 5). In contrast, even after pre-incubation with PKA and ATP, the T389A-MAST3-HA mutant was only slightly less active than WT MAST3-HA. The conclusion from this in vitro experiment was that T389 is phosphorylated by PKA, the T389D mutation acts as a phospho-mimetic to inhibit MAST3 activity, while the T389A mutant largely prevents the effect of PKA. This data was originally shown as a supplementary figure but we have moved it into the main figures (now Figure 5).

We then analyzed the effects of mutations of four different phosphorylation sites (T389, S512, T628, and S747) on MAST3 activity in intact Hek cells. Sites were individually mutated to either alanine (non-phosphorylatable) or aspartate (phospho-mimetic). The T389D mutant, consistent with the in vitro assay, exhibited very low activity as demonstrated by the very low level of S46 phosphorylation (compare lanes 1 and 3 in the blot in Figure 5, or the first and third black bar in Figure 5). The effects of the S512D and T628D mutations were similar to each other, both displayed reduced MAST3 activity (see Figure 6), but this was not as much as the T389D mutation. The S747D mutant exhibited activity similar to the WT protein. With respect to the alanine mutations, the in vitro and intact cell experiments indicated that phosphorylation of Thr389 had the most robust effect on MAST3 activity. Mutation of Thr389 to alanine reduced the efficacy of forskolin to reduce Ser46 phosphorylation (Figure 5). In contrast, the Ser512 or Thr628 to alanine mutants had no effect on the ability of forskolin to reduce Ser46 phosphorylation (Figure 6), presumably because of the predominant effect of Thr389 phosphorylation. We have clarified the description of these experiments in the revised manuscript.

*8) The ability of Ser88 phosphorylated ARPP-16 to act as a dominant inhibitor of Ser64 phosphorylated ARPP-16 is intriguing. This suggests that the two ARPP-16 proteins might compete for binding to PP2A and that pSer88 ARP-16 has a higher affinity for PP2A than pSer64 ARPP-16. Alternatively, the binding may be non-competitive and pSer88 ARPP-16 may have dominant activity. Other mechanisms are also possible. Studies to evaluate mechanism are required, including the determination of binding affinity and binding competition.*

We agree that this observation is intriguing, although as noted above and in the revised manuscript, based on the additional molecular modeling its functional significance may not be as great as the other two layers of regulation. We also agree that some sort of quantitative method that would determine binding affinity and mechanism is warranted. To achieve this we considered several options. (1) Kinetic analyses based on PP2A activity assays; (2) Direct binding experiments using ITC, SPR, or some sort of solution binding experiment combined with, for example, Scatchard analysis. While we have previously found that the A subunit of PP2A can bind directly to ARPP-16, it is likely that phospho-Ser46 binds to active site of the C subunit. Moreover, inhibition by phospho-ARPP-16 is not observed with the AC dimer (see (Andrade et al., 2017)), and specificity is exhibited for either the Bα or B56δ isoforms (Mochida et al., 2009; Mochida et al., 2010; Andrade et al., 2017). For these various reasons, we would need reasonable amounts of a purified heterotrimeric complex to carry out detailed analyses. This would go far beyond our current capabilities that are currently done with small-scale, partially purified, Hek cell preparations that exhibit experiment-to-experiment variation. We have begun to establish the resources to purify the required proteins, but we were not able to do this in the limited time available.